# Integrating 3D genomic and epigenomic data to enhance target gene discovery and drug repurposing in transcriptome-wide association studies

Chachrit Khunsriraksakul [1,2], Daniel McGuire [2,3], Renan Sauteraud[2,3], Fang Chen[2,3], Lina Yang[2,3], Lida Wang [2,3], Jordan Hughey[1,2], Scott Eckert[1,2], J. Dylan Weissenkampen[2,3], Ganesh Shenoy[4], Olivia Marx[5], Laura Carrel[6], Bibo Jiang [3✉] & Dajiang J. Liu [1,2,3✉]

Transcriptome-wide association studies (TWAS) are popular approaches to test for association between imputed gene expression levels and traits of interest. Here, we propose an integrative method PUMICE (Prediction Using Models Informed by Chromatin conformations and Epigenomics) to integrate 3D genomic and epigenomic data with expression quantitative trait loci (eQTL) to more accurately predict gene expressions. PUMICE helps define and prioritize regions that harbor cis-regulatory variants, which outperforms competing methods. We further describe an extension to our method PUMICE +, which jointly combines TWAS results from single- and multi-tissue models. Across 79 traits, PUMICE + identifies 22% more independent novel genes and increases median chi-square statistics values at known loci by 35% compared to the second-best method, as well as achieves the narrowest credible interval size. Lastly, we perform computational drug repurposing and confirm that PUMICE + outperforms other TWAS methods.

[1] Bioinformatics and Genomics Graduate Program, Pennsylvania State University College of Medicine, Hershey, PA 17033, USA. [2] Institute for Personalized Medicine, Pennsylvania State University College of Medicine, Hershey, PA 17033, USA. [3] Department of Public Health Sciences, Pennsylvania State University College of Medicine, Hershey, PA 17033, USA. [4] Department of Neurosurgery, Pennsylvania State University College of Medicine, Hershey, PA 17033, USA. [5] Biomedical Science Program, Pennsylvania State University College of Medicine, Hershey, PA 17033, USA. [6] Department of Biochemistry and Molecular Biology, Pennsylvania State University College of Medicine, Hershey, PA 17033, USA. ✉email: bjiang@phs.psu.edu; dajiang.liu@psu.edu

With the rapid decline in high-throughput genotyping costs, GWAS has been widely used to search for genetic risk factors predisposing individuals to complex diseases or traits. These studies to date have identified tens of thousands of single nucleotide polymorphisms (SNPs) associated with complex phenotypes, yet many of these SNPs are noncoding, which are challenging to interpret their biological functions. It has become clear that the integration of multi-omics data is necessary for elucidating the biological functions and clinical consequences of noncoding variants[1].

Recently, a novel approach named transcriptome-wide association studies (TWAS), has been proposed and gained popularity as an alternative way to perform gene-based association analyses. Briefly, TWAS derives the gene expression prediction models from datasets with matched genotypes and gene expression data (e.g., Genotype-Tissue Expression project (GTEx)[2], Depression Gene Network (DGN)[3], Common Mind Consortium (CMC)[4], Genetic European Variation in Disease (GEUVADIS)[5]). Using these models, gene expression levels can be imputed in a GWAS dataset and then tested for associations with complex traits, in order to identify genetically-regulated target genes. To date, numerous TWAS methods have been developed, with each method using different specifications of regression models to construct gene expression prediction models[6–10].

Since the invention of chromosome conformation capture technology, the 3D genome structure has provided important novel insights into cis-regulatory elements[11]. Specifically, chromosomes occupy distinct positions within the nucleus, called chromosome territories, which can be partitioned into chromosomal compartments and further partitioned into domains and loops[12]. Importantly, it has been shown that genomic segments within the same domain have a higher frequency of interactions with each other than they do with neighboring regions[13,14]. The 3D organization of the genome is closely related to numerous key biological functions including gene expression, DNA repair, and DNA replication regulations[15]. In parallel, epigenetic processes, including DNA methylation and histone modification, are thought to influence gene expression[16]. Genetic variants which overlap with key annotation tracks (e.g., enhancers) are more likely to be functionally relevant and should be prioritized in the development of a gene expression prediction model[17]. EpiXcan[9] is the first method that attempts to incorporate epigenomic data to improve prediction accuracy. Specifically, EpiXcan generates priors from Roadmap chromHMM annotation[18] and uses quadratic Bezier functions to rescale SNP priors to penalty factors which are used in a weighted elastic net regression. However, it is computationally expensive to obtain a rescaling equation and currently the authors only used 8 representative genes to pick an optimal Bezier function. To our knowledge, none of the TWAS methods to date integrates 3D genomic data in modeling gene expression levels.

Here, we showcase the power of utilizing both 3D genomic and epigenomic data to more accurately model gene expression and more powerfully conduct TWAS, via a new method which we call PUMICE (Prediction Using Models Informed by Chromatin conformations and Epigenomics). PUMICE outperforms existing single-tissue and multi-tissue methods in terms of the predictive accuracy of gene expression levels, the power for identifying associated genes, and the fine-mapping resolution in the analyses of 79 complex traits/diseases. PUMICE can be further improved by combining with multi-tissue TWAS method UTMOST (which we call PUMICE+), as multi-tissue methods offer complementary information. We also performed computational drug repurposing using our TWAS results and identified clinically-relevant small molecule compounds that may be repurposed for treating immune-related traits, COVID-19-related outcomes, and other relevant disease outcomes.

## Results

**Overview of PUMICE Model for Gene Expression Levels.** Here we outline the PUMICE method for predicting gene expression levels using eQTL effects, epigenomic annotation, and 3D genomes. More detailed descriptions can be found in Methods. PUMICE utilizes epigenomic information to prioritize essential genetic variants that carry important functional roles. Specifically, we utilize four broadly available epigenomic annotation tracks, including H3K27ac mark, H3K4me3 mark, DNase hypersensitive mark, and CTCF mark from ENCODE database. Variants which overlap these annotation tracks are deemed "essential" genetic variants. We use $\mathbf{X_1}$ to represent the set of essential genetic variants, $\mathbf{X_2}$ to denote "non-essential" genetic variants that do not overlap annotation tracks, and $\mathbf{E}$ to represent the vector of gene expression levels across multiple individuals.

The prediction model seeks to estimate the weights $\boldsymbol{\beta_1}$ and $\boldsymbol{\beta_2}$ by minimizing the following objective function:

$$L(\boldsymbol{\beta_1}, \boldsymbol{\beta_2}; \lambda, \phi) = \|\mathbf{E} - \mathbf{X_1}\boldsymbol{\beta_1} - \mathbf{X_2}\boldsymbol{\beta_2}\|_2^2 + \frac{1}{2} \times \frac{\lambda}{2}(\phi\|\boldsymbol{\beta_1}\|_2^2 + \|\boldsymbol{\beta_2}\|_2^2)$$
$$+ \frac{1}{2}\lambda(\phi\|\boldsymbol{\beta_1}\|_1^1 + \|\boldsymbol{\beta_2}\|_1^1)$$

$$(1)$$

The model utilizes both ridge and lasso penalties, as decided by the tuning parameter $\lambda$. $\phi$ is another tuning parameter that controls the penalty on the essential predictors relative to the non-essential predictors[19]. We assume $\phi \leq 1$ so that essential predictors are penalized no more than the non-essential predictors. In our implementation, we use nested cross-validation to select an optimal value of $\phi$ between 0 and 1, as we do not know a priori the extent to which essential and non-essential variants should be differentially penalized. We also considered different choices for regions that harbor cis-regulatory variants as another tuning parameter (denoted as $w$), which includes the ones defined by linear windows surrounding gene start and end sites as well as by 3D genomics informed regions (i.e., loop, TAD, domain, and promoter capture Hi-C (pcHi-C) region). We utilize nested cross-validation to select the optimal combinations of tuning parameters $\lambda$, $\phi$, and $w$.

**Simulation studies.** We conducted extensive simulations to evaluate different methods. We considered scenarios with different proportions of causal variants, scenarios where regulatory regions are defined by linear windows from gene start and end sites or defined by 3D genomes, as well as scenarios where gene expression heritability are enriched in essential variants overlapping epigenomic annotation tracks and where heritability is evenly distributed across causal variants. To evaluate multi-tissue methods, we also simulated gene expressions of a causal tissue, varying number of correlated tissues (where the eQTL effect sizes are correlated with a causal tissue), and other noncorrelated tissues (where the eQTL effect sizes are not correlated with a causal tissue). Detailed descriptions of the simulation study design can be found in Methods.

When compared to all other single-tissue TWAS methods, we observed that PUMICE attained the highest imputation accuracy, power, and well-controlled type I error (Fig. 1a, b, Supplementary Fig. 1a, Supplementary Fig. 2, and Supplementary Data 1). Importantly, the gains in imputation accuracy (13% on average), the number of significant models (average of 68%) and power (average of 63%) by PUMICE were particularly high when training sample size is small ($n = 100$).

Comparing to methods that do not impose sparsity (i.e., FUSION and TIGAR), we observed bigger gains of TWAS power by PUMICE when number of causal SNPs is smaller or when

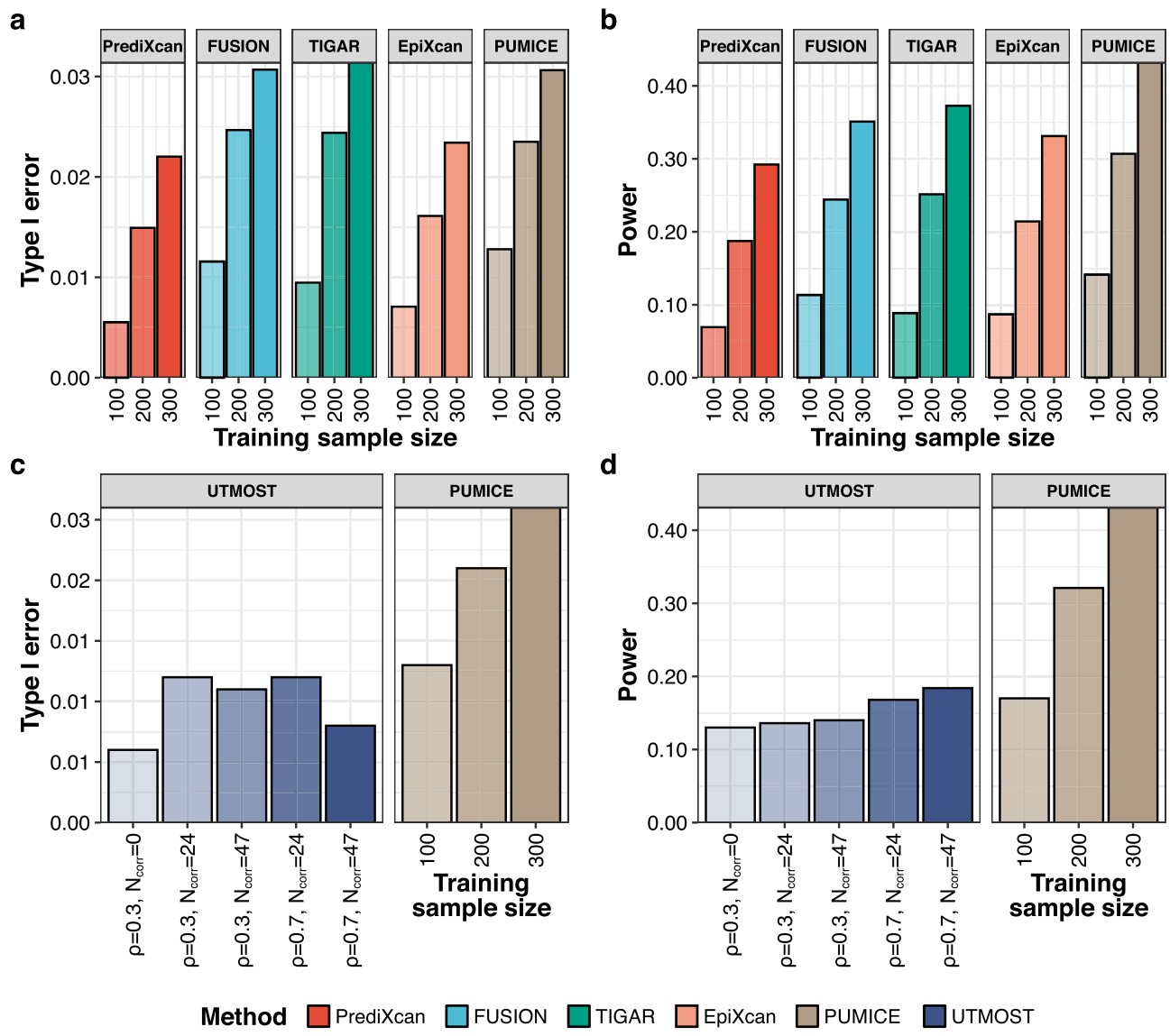

**Fig. 1 Simulation studies comparing the performance of PUMICE to other TWAS methods.** Panels (**a**, **b**) illustrates the comparison of PUMICE to other single-tissue TWAS methods for type I error (**a**) and power (**b**). Panels (**c**, **d**) illustrates the comparison of PUMICE to multi-tissue TWAS method (UTMOST) for type I error (**c**) and power (**d**). For UTMOST, we evaluate its performance across different combinations of genetic correlation between causal and correlated tissues ($\rho$) and number of correlated tissues ($N_{corr}$). Shadings represent different training sample sizes used to train gene expression prediction models for single-tissue TWAS methods and $\rho/N_{corr}$ combinations for multi-tissue TWAS method.

expression heritability increased. In contrast, for methods imposing sparsity (i.e., PrediXcan and EpiXcan), we observed an opposite trend where the power gain by PUMICE increases with the number of causal variants or when expression heritability decreased. Moreover, the power advantages by PUMICE over other methods increased when the proportion of causal SNPs in epigenomic regions increased, or when there is a stronger enrichment of heritability among essential variants, or when training sample size decreased.

In comparison to multi-tissue TWAS method (UTMOST), we found that PUMICE outperformed UTMOST in terms of external prediction accuracy and TWAS power when training sample size is high (i.e., 200 or 300), when genetic correlation between causal and correlated tissues is low (i.e., 0.3), and when number of correlated tissues (i.e., 0 or 24) is low (Fig. 1c, d, Supplementary Fig. 1b). Only when training sample size is very small ($n = 100$) did UTMOST outperform PUMICE in terms of prediction accuracy (median Spearman's correlations of 0.17 vs. 0.13).

However, in this case, UTMOST's TWAS power is also only slightly better than PUMICE (power of UTMOST vs. PUMICE is 0.18 vs. 0.17).

**Prediction performance of PUMICE and other TWAS methods in GTEx tissues.** We compared PUMICE against single-tissue TWAS methods (PrediXcan[6], FUSION[7], TIGAR[8], and EpiXcan[9]) and a multi-tissue TWAS method (UTMOST[10]) using cross validation. Here, the imputation accuracy of each method was measured by the median value of Spearman's rank correlation coefficient (r). More details on how cross-validation was performed can be found in the Methods.

Compared to PrediXcan, PUMICE achieved higher imputation accuracy and identified more significant genes across 48 GTEx tissues (Fig. 2, Supplementary Fig. 3, and Supplementary Data 2-3). On average, the imputation accuracy was improved by 28% and the number of significant prediction models was increased by 51% across all tissues. The improvement in prediction performance was

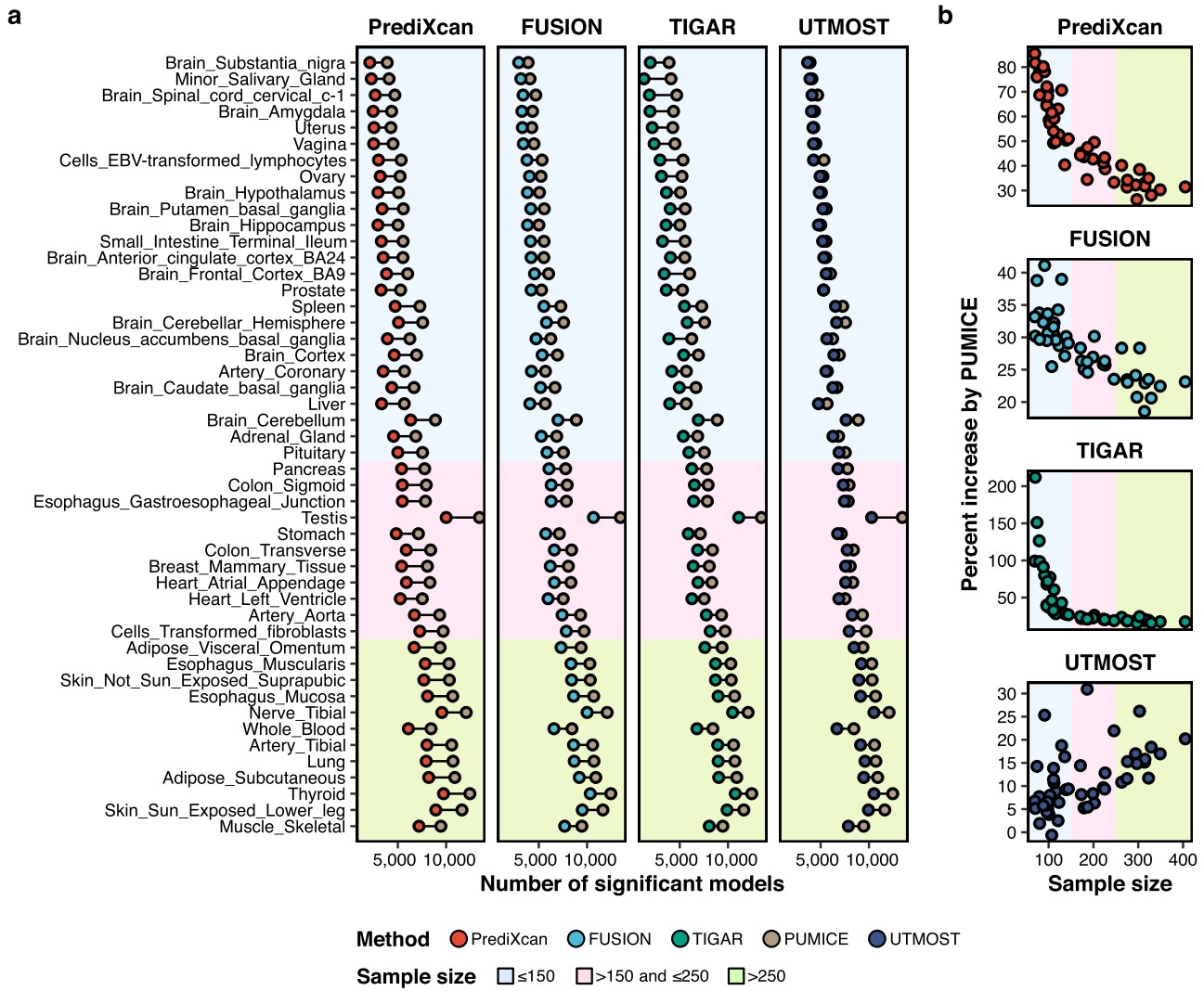

**Fig. 2 Comparison of PUMICE gene expression prediction models to other TWAS methods.** In panel (**a**), we compare the number of PUMICE significant models to other TWAS methods, including PrediXcan, FUSION, TIGAR, and UTMOST, across 48 GTEx tissues. Across all scenarios, PUMICE achieves higher number of significant models than those of other single-tissue TWAS methods. In comparison to UTMOST, PUMICE achieves comparable number of models in small sample size tissues, but achieves higher number of models in large sample size tissues. In panel (**b**), we illustrate the percent increase in significant models between PUMICE and other TWAS methods. Comparing to other single tissue TWAS methods, percent increase of models gets larger for smaller training sample size. In comparison to UTMOST, percent gain of models gets larger for larger training sample sizes.

particularly high in tissues with small sample sizes ($N \leq 150$), where we observe an average gain of 36% in imputation accuracy and 63% in number of models. Similar trends were observed when comparing PUMICE to FUSION (35% gain in average imputation accuracy and 28% gain in the number of significant models) and when comparing PUMICE to TIGAR (42% gain in average imputation accuracy and 43% gain in the number of significant models) (Fig. 2, Supplementary Figs. 4 and 5, and Supplementary Data 2-3).

In comparison to EpiXcan, PUMICE, on average, improved the imputation accuracy by 27% and the number of significant prediction models by 41% (Supplementary Fig. 6 and Supplementary Data 2-3). Highest gain was also seen in tissues with small sample sizes. It should be noted that only 8 GTEx tissues were compared due to the limited availability of the EpiXcan models.

Compared to UTMOST, PUMICE achieved higher imputation accuracy (by an average of 14%) and higher number of significantly imputed genes (by an average of 11%) across 48 GTEx tissues (Fig. 2, Supplementary Fig. 7, and Supplementary

Data 2-3). The improvement in prediction performance was particularly high in tissue with large sample sizes ($n>250$) (average gain of 21% in imputation accuracy and 16% in number of models). For tissues with smaller sample sizes (e.g., Brain substantia nigra, $n = 69$), we observed a comparable performance with UTMOST.

To further quantify the similarity/difference of each TWAS method, we compared the significant models being identified. On average, 92% and 91% of the PrediXcan's and EpiXcan's significant models were significant in PUMICE, but only 62% and 66% of PUMICE models were significant in PrediXcan and EpiXcan, which illustrated the improved power for PUMICE. Interestingly, PUMICE and UTMOST usually identified a different set of significant models (average of 55% overlap), demonstrating the fundamental difference between single-tissue and multi-tissue TWAS methods. As we will show in the TWAS section, models arising from multi-tissue methods tend to be copied across tissues, as the group lasso algorithm used in UTMOST preferentially select the same predictors across tissues[10]. Although they may improve the prediction accuracy

in small sample size tissues, they did not properly capture tissue specificity and did not help with gene discovery and fine mapping.

**Prediction performance of PUMICE and other TWAS methods in external datasets.** We compared the prediction performance of PUMICE with other TWAS methods, including PrediXcan[6], FUSION[7], TIGAR[8], EpiXcan[9], and UTMOST[10], in the independent datasets. Specifically, we focused on 3 GTEx tissues with matching external datasets (i.e. whole blood from GTEx and DGN[3], brain frontal cortex BA9 from GTEx and CMC[4], and lymphoblastoid cell lines from GTEx and GEUVADIS[5]). We derived gene expression prediction models using GTEx data and compared prediction performance using independent datasets.

Across three different tissues, PUMICE significantly improved the prediction performance in comparison to all other single-tissue TWAS methods (Supplementary Fig. 8 and Supplementary Data 4). To further confirm that PUMICE performance in comparison to EpiXcan was not solely due to the difference in epigenomic annotation being used, we trained PUMICE with Roadmap ChromHMM annotations, which EpiXcan also uses. We found that PUMICE's external prediction performance was still 6.81% higher than that of EpiXcan (in comparison to 8.17% improvement when PUMICE used SCREEN annotations) (Supplementary Data 5). This further illustrated the flexibility of PUMICE to utilize various sources of functional annotation data. When compared with UTMOST, PUMICE performed better in whole blood tissue and worse in the other two tissues (Supplementary Data 4). We attribute this finding to a relatively small sample size of GTEx brain frontal cortex BA9 ($n = 103$) and lymphoblastoid cell line ($n = 91$) which limits the power of single-tissue-based TWAS methods.

Multi-tissue TWAS methods often require measured gene expression from multiple tissues in the same individuals. Its main advantage is limited to tissues of small sample sizes. They often perform worse than single-tissue methods in GTEx tissues with large sample sizes (i.e., $n > 250$). In many scenarios, non-GTEx dataset from some tissues can be bigger in size. Single tissue prediction models trained in these larger non-GTEx datasets can yield much improved accuracy than multi-tissue methods trained in GTEx tissues, which makes the single tissue prediction models a much-preferred choice in downstream analysis. For example, we trained PUMICE in larger training datasets (i.e., DGN ($n = 873$), CMC ($n = 380$), and GEUVADIS ($n = 358$)) and evaluated these models in the matched GTEx tissues. In agreement with our simulation results, we found that PUMICE's prediction performance and TWAS power significantly improved in comparison to that of UTMOST trained in GTEx (Supplementary Data 6-7). Examples of well-imputed genes are shown in Fig. 3, illustrating improved prediction performance in PUMICE over other TWAS methods.

**Characteristics of PUMICE prediction models.** PUMICE uses a data-driven approach to determine the combination of tuning parameters $\phi$ and $w$ that may best predict expression levels. We first investigated frequency of each combination of $w$ and $\phi$ among the selected models in cross-validation evaluations across 48 GTEx tissues. Among all significant genes, PUMICE selected 3D genome informed regions ($w$ = loop, TAD, domain, pcHi-C) 52% of the time. Among the best models, the most frequent choice for $w$ was 250 kb window surrounding the gene start and end site (35% of the time) (Fig. 4a and Supplementary Data 8). This contrasts the common choice of a 1 million base-pair window surrounding each gene frequently used by alternative methods. The selected predictors using PUMICE are clearly

concentrated in smaller windows as defined by linear distance from genes and in regions defined by 3D genome organizational structures.

Leveraging epigenomic data, PUMICE is able to prioritize essential predictors by assigning them a smaller penalty. We noted $\phi = \frac{1}{6}$ as the most frequent choice (22%) for PUMICE prediction models, which assigns a much smaller penalty term to essential predictors (Fig. 4b and Supplementary Data 9). This indicates that the inclusion of essential predictors as determined by epigenomic annotations tends to improve prediction accuracy.

Finally, our results showed that PUMICE produced compact and parsimonious models which include relatively small number of SNPs with non-zero weight. In our analysis, the median number of SNPs with non-zero weights for PUMICE was 13 SNPs/model, in comparison to 17, 24, and 73 SNPs/model for EpiXcan, PrediXcan, and UTMOST, respectively (Fig. 4c). As expected, for all methods, SNPs with non-zero weights are enriched near gene transcription start and end sites and PUMICE shows the strongest enrichment (Fig. 4d).

**TWAS analysis of GWAS summary statistics from 79 human complex traits.** We applied PrediXcan, FUSION, TIGAR, PUMICE, and UTMOST prediction models from 48 GTEx tissues to 79 complex traits (Supplementary Data 10) and performed TWAS. Overall, PUMICE outperforms all other methods (i.e., PrediXcan, FUSION, TIGAR, and UTMOST), in terms of 1) the most extreme departure from the null at the tail of the distribution in quantile-quantile plots, 2) the number of unique genes identified across tissue types, where gene trait associations identified in multiple tissues are counted only once, 3) the number of independent genes identified, 4) the number of novel genes identified, and 5) the highest median chi-square value at known loci (Supplementary Fig. 9 and Fig. 5). It should be noted that EpiXcan was not included in the analysis because too few EpiXcan models were available.

Although UTMOST identified more gene x trait associations than PUMICE, most of these associations tend to be copied across different tissue types, due to the group lasso algorithm used in UTMOST that preferentially select predictors shared across tissues. UTMOST performed worse than PUMICE in other comparisons as shown above, including the number of unique gene x trait associations, independent gene counts, and novel gene counts.

PUMICE can be further improved by combining it with a multi-tissue method. It has been observed previously that single tissue-based (e.g., PrediXcan) and multi-tissue-based (e.g., UTMOST) TWAS methods often identified distinct gene x trait associations[20]. We applied Cauchy combination test to combine the results of different single-tissue TWAS methods with UTMOST (denoted as + method). For example, PUMICE+ refers to the Cauchy combination test of PUMICE and UTMOST TWAS results. PUMICE+ further improved the performance of PUMICE and exhibited a greater advantage over all other methods. Specifically,

1) In quantile-quantile plots of $p$-values of each TWAS, PUMICE+ showed the most extreme departure from the null at the tail of the distribution (Supplementary Fig. 9b).
2) PUMICE+ identified the largest number of significant gene x trait associations (Fig. 5a and Supplementary Data 11), i.e., PUMICE+ identified 8% more significant gene x trait associations than the second-best method (FUSION+).
3) PUMICE+ identified the largest number of unique significant gene x trait associations across different tissues (Fig. 5b and Supplementary Data 12). PUMICE+ identified 24% more unique significant gene x trait associations than the second-best method (FUSION+).

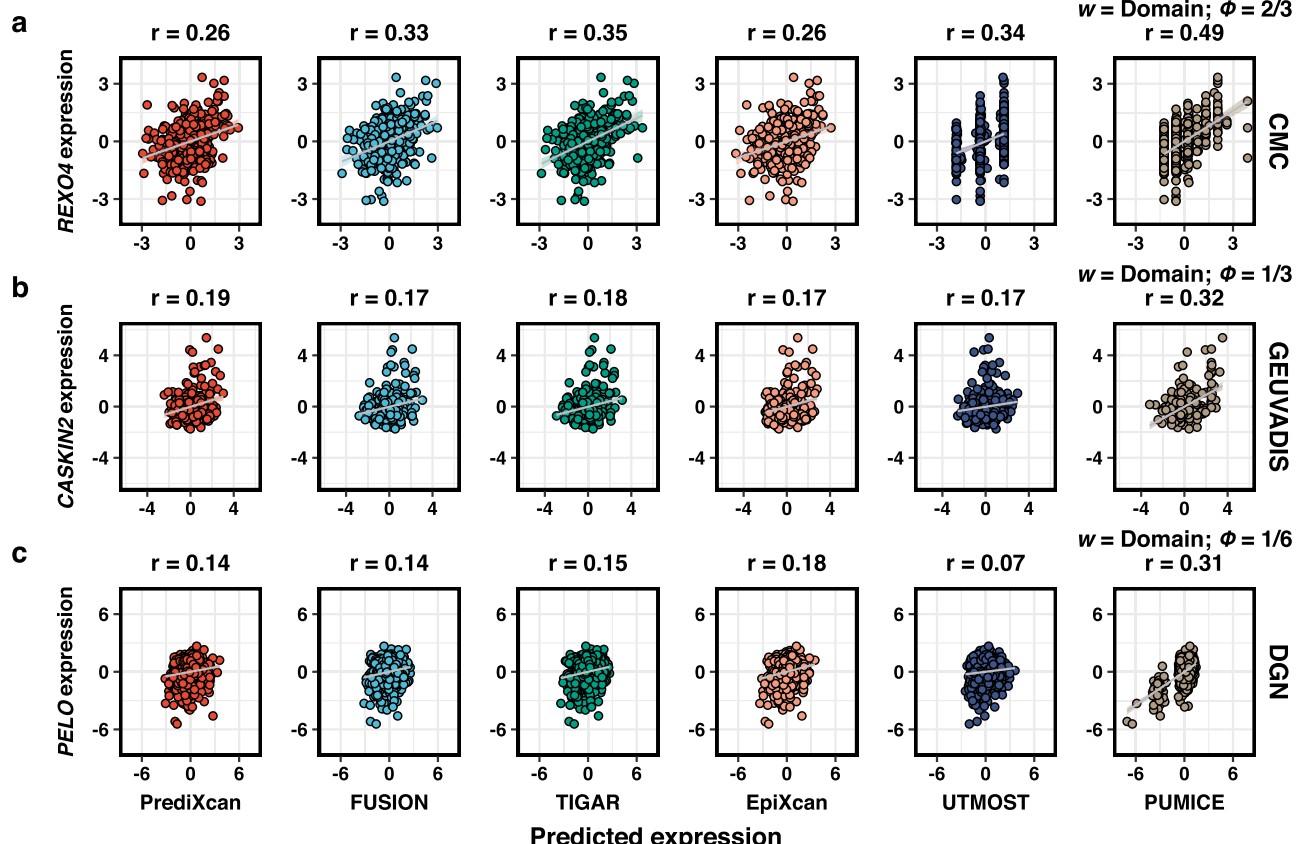

**Fig. 3 Examples of well-imputed genes unique to PUMICE.** Panel (**a**) displays prediction performance of *REXO4* gene in CMC cohort. Panel (**b**) shows prediction performance of *CASKIN2* gene in GEUVADIS cohort. Panel (**c**) shows prediction performance of *PELO* gene in DGN cohort. We show the selected window (*w*) and penalty factor ($\phi$) associated with each PUMICE's prediction model. Error bands represent 95% confidence intervals.

4) PUMICE+ identified the largest number of independent gene x trait associations. Here, we systematically pruned genes, and only counted the most significant genes in each 1 Mb window as independent genes. In comparison to second-best method (PrediXcan+), PUMICE+ identified 15% more independent gene x trait associations (Fig. 5c and Supplementary Data 13).

5) PUMICE+ identified the largest number of independent and novel gene x trait associations outside GWAS loci. Here we defined GWAS loci as a ± 1 Mb region surrounding the sentinel variants. In comparison to second-best method (PrediXcan+), PUMICE+ identified 22% more independent and novel gene x trait associations located outside the GWAS loci (Fig. 5d and Supplementary Data 14).

6) PUMICE+ also attained the largest median chi-square value at known loci, which can be used as an empirical estimate of power[21]. Here, we use MAGMA-prioritized genes as proxy for known genes[22]. Among all methods compared, PUMICE+ achieved the largest median value of chi-square test statistics at MAGMA-prioritized genes (Fig. 5e). PUMICE+ increased the median chi-square statistics values at known loci by 35%, compared to the second-best method (PrediXcan+).

**TWAS-based computational drug repurposing.** We performed a computational drug repurposing (CDR) using Connectivity Map (CMap) approach to identify bioactive small molecules capable of reversing the expression profile of clinically relevant trait-associated genes (Supplementary Data 15). CMap database consists of a reference collection of ~1 million gene-expression profiles from cultured human cells treated with 50,000 bioactive small molecules[23,24]. In order to relate disease to drug-induced state, CMap calculates a $\tau$-score to assess the correlation between a query signature (a Z-score measuring how gene expression is associated with disease status) and a reference profile (measuring how a given component modifies gene expression). A negative connectivity score emphasizes an inverse similarity between a query signature and a reference profile[25], indicating the potential utility of the identified molecule to normalize trait-associated gene expression profile.

To objectively measure the CDR performance across different TWAS methods, we compared the CDR predictions with known drug indications for 23 diseases (Methods, Fig. 6, and Supplementary Data 15-16). We did not perform drug repurposing for quantitative outcomes, which are based on normal trait variations in the general population, as it is difficult to interpret the results for those quantitative traits. PUMICE+ achieved the lowest median CMap score ($\tau$) ($\tau = -77.99$), which shows that it identifies putative target genes that are most consistent with target genes of approved drugs.

PUMICE+ identified novel drug classes, including VEGFR inhibitor for asthma, proteasome inhibitor for inflammatory bowel disease, Hsp90 inhibitor for rheumatoid arthritis, JAK inhibitor for vitiligo, SRC inhibitor for coronary artery disease, mTOR inhibitor for schizophrenia, and IGF-1 inhibitor for bipolar disorder, that may be worth investigating further. VEGF has an important role in airway inflammation during acute asthma and administration of VEGFR inhibitor in murine model was shown to reduce asthmatic symptoms[26,27]. Bortezomib, a proteasome inhibitor, can ameliorate the disease activity of

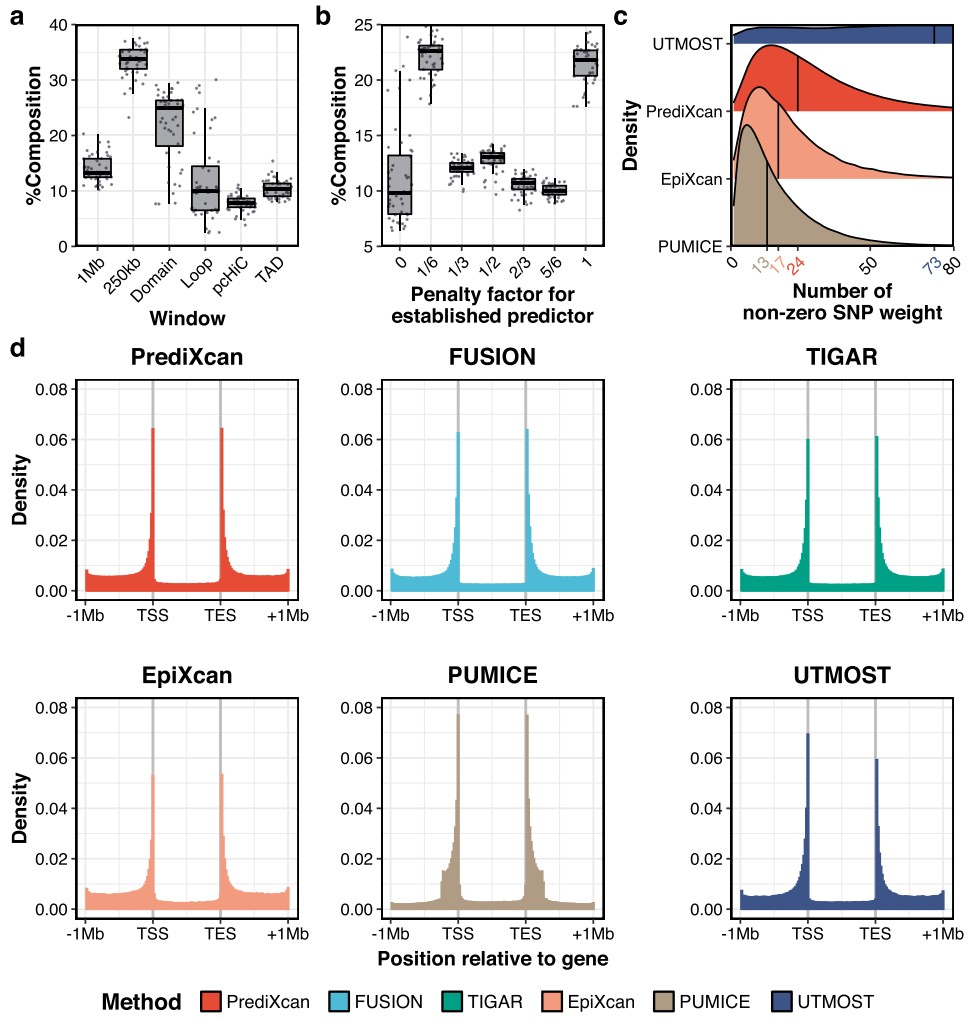

**Fig. 4 Characteristics of GTEx gene expression prediction models.** In panel (**a**), we illustrate the proportions of different window sizes *w* among selected PUMICE models. Each boxplot is derived from the percent window composition of 48 GTEx tissues. In panel (**b**), we show the proportion of different values of tuning parameter $\phi$ among selected PUMICE models. $\phi$ is the tuning parameter that reduces the $L_1$ and $L_2$ penalties for essential predictors that overlap with ENCODE annotations. Each boxplot is derived from the percent penalty factor composition of 48 GTEx tissues. Minima and maxima values (excluding outliers) are represented by the lower- and upper-bound of the whiskers. Median value is represented by the bolded line in the middle. First and third quartiles are represented by the lower- and upper-bound of the box. In panel (**c**), we show the distribution of the number of SNPs with non-zero weights in gene expression prediction models across different TWAS methods. Vertical line represents median number of SNPs with non-zero weights. PUMICE models have the lowest median number of SNPs with non-zero weights ($n = 13$), while UTMOST models have the highest median number of SNPs with non-zero weights ($n = 73$). In panels (**d**), we plot the distribution of the locations of SNPs with non-zero weights (for PrediXcan, EpiXcan, PUMICE, and UTMOST) or top 100 SNPs with highest weights (for FUSION and TIGAR). Variant counts are plotted against their locations relative to 5' gene transcription start site (TSS) and 3' gene transcription end site (TES) across different TWAS methods.

murine colitis[28]. Hsp90 inhibitors have been shown to ameliorate various autoimmune and inflammatory diseases, including rheumatoid arthritis, in rodents[29–33]. Treatment with ruxolitinib, a JAK inhibitor, was associated with re-pigmentation of vitiligo lesions according to the result from a recent randomized, phase 2 clinical trial[34]. SRC inhibitor may have a potential role for treating cardiovascular diseases[35]. There is growing evidence that mTOR dysfunction may underlie a variety of psychiatric syndromes, including schizophrenia[36]. Lastly, peripheral IGF-1 level is increased in patients with bipolar disorder[37]. Role of IGF-1 inhibitor in bipolar disorder may be worth looking into.

**Fine-mapping of TWAS results**. We performed fine-mapping of TWAS results across 79 traits and 48 GTEx tissues. Using FOCUS[38], we found that PUMICE achieved a significantly narrower average 90%-credible interval (average of 1.7) in

comparison to all other TWAS methods (Supplementary Data 17). This is likely because PUMICE uses data-driven methods to select biologically informative predictors, which reduces correlations between imputed gene expressions. Since FOCUS could not perform fine-mapping for methods based on Cauchy-combined *p*-values, we applied our fine-mapping method and demonstrated that PUMICE+ also achieved the narrowest 90%-credible interval (average of 1.4) (Supplementary Data 17). Comparing the performance between FOCUS and our fine-mapping method, we found that 78% of the credible sets on average contained the same fine-mapped genes, illustrating an agreement between FOCUS and our fine-mapping method (Supplementary Data 17).

**PUMICE+ in-silico follow-up and drug repurposing analysis for COVID-19**. We conducted in-silico follow-up of four

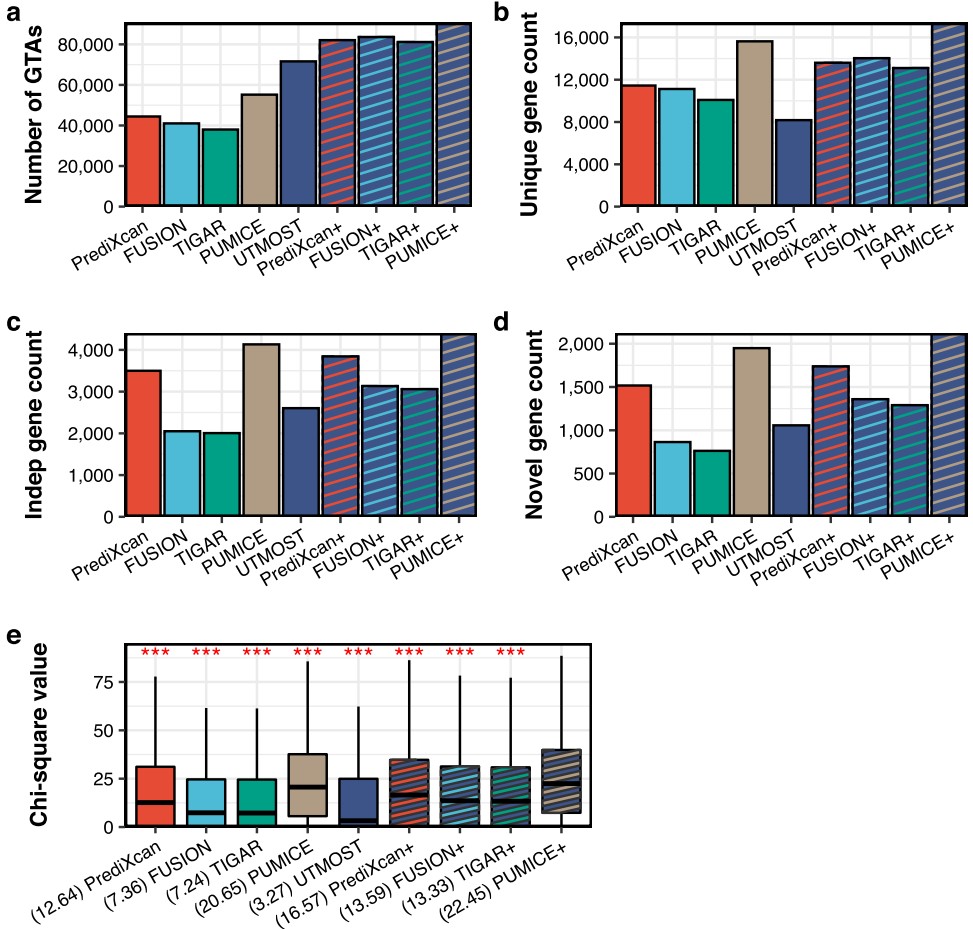

**Fig. 5 PUMICE+ identifies the largest number of gene x trait associations and novel associations across 48 GTEx tissues using GWAS summary-level statistics of 79 traits.** Panel (**a**) displays the total number of significant gene x trait associations by each method. Panel (**b**) shows the number of unique significant gene x trait associations. Gene x trait associations identified in multiple tissues are counted only once. Panel (**c**) shows the number of independent gene counts. Multiple significant genes within 1 Mb window are counted only once. Panel (**d**) shows the number of independent novel genes that are outside 1 Mb window on either side of GWAS sentinel variant. PUMICE+ identifies the highest number of gene x trait associations, unique gene count, independent gene count, and novel gene count in comparison to all other methods. Panel (**e**) displays the distribution of chi-square test statistics at MAGMA-prioritized genes. Median value is denoted in the parentheses. PUMICE+ achieved the largest median value of chi-square values (22.45). *P*-value is based on the comparison between PUMICE+ and other method using one-sided median test. *** denotes significant differences with *p* < 0.001. One-sided *p*-values are provided in the source data file. Each boxplot is derived from the chi-square values across 12,546 MAGMA-prioritized genes. Minima and maxima values (excluding outliers) are represented by the lower- and upper-bound of the whiskers. Median value is represented by the bold line in the middle. First and third quartiles are represented by the lower- and upper-bound of the box.

COVID-19 GWAS datasets from the COVID-19 Host Genetic Initiative to learn more about the genetic determinants of COVID-19 susceptibility, severity, and outcomes[39]. Classically, COVID-19 is characterized by the symptoms of viral pneumonia; however, it has been shown that COVID-19 may cause damage to multiple organs/organ systems including heart, liver, kidney, blood, and immune system[40–42]. As a baseline comparison, we applied models across 48 GTEx tissues of each TWAS method to the COVID-19 datasets and confirmed through quantile-quantile plots that PUMICE and PUMICE+ showed the most extreme departure from the null at the tail end of the distribution (Supplementary Fig. 9c, d).

Using TWAS, we investigated disease-associated genes and found multiple hits with potential biological connections to the pathogenesis of the disease (Fig. 7).

A few genes, including *SLC6A20*, *ABO*, *ELF5*, *OAS1/3*, *MAPT*, and *IL10RB*, were identified across multiple COVID GWAS datasets. *SLC6A20* encodes an amino acid transporter that was shown to interact with *ACE2*[43]. ABO blood group has been

shown to be associated with the COVID-19 outcomes[44]. Specifically, group A is conferring a higher susceptibility to infection and propensity to severe disease. *ELF5* was previously shown to be associated with poor SARS-CoV-2 prognosis[45]. *OAS1/3* encodes oligoadenylate synthetases which can activate ribonuclease L, an enzyme that degrades intracellular double-stranded RNA[46]. *MAPT* encodes microtubule-associated protein tau. Tau protein was shown to bind to SARS-CoV-2 Spike S1 protein receptor-binding domain and accelerate the aggregation of pathological amyloid proteins in the brain, resulting in neurodegeneration. Importantly, neurologic symptoms including neurodegeneration can be observed post-COVID infection[47]. In vitro *IL10RB* overexpression is associated with increased viral load and activation of immune-related pathways[48].

For GWAS of very severe respiratory confirmed COVID vs. population (COVID-A2 dataset), we identified a few biologically-relevant genes, including *GBA*, *TRIM23*, *ATP11A*, and *TYK2* (Fig. 7a). *GBA* is associated with Parkinson's disease, and a few cases have been reported about the development of acute or

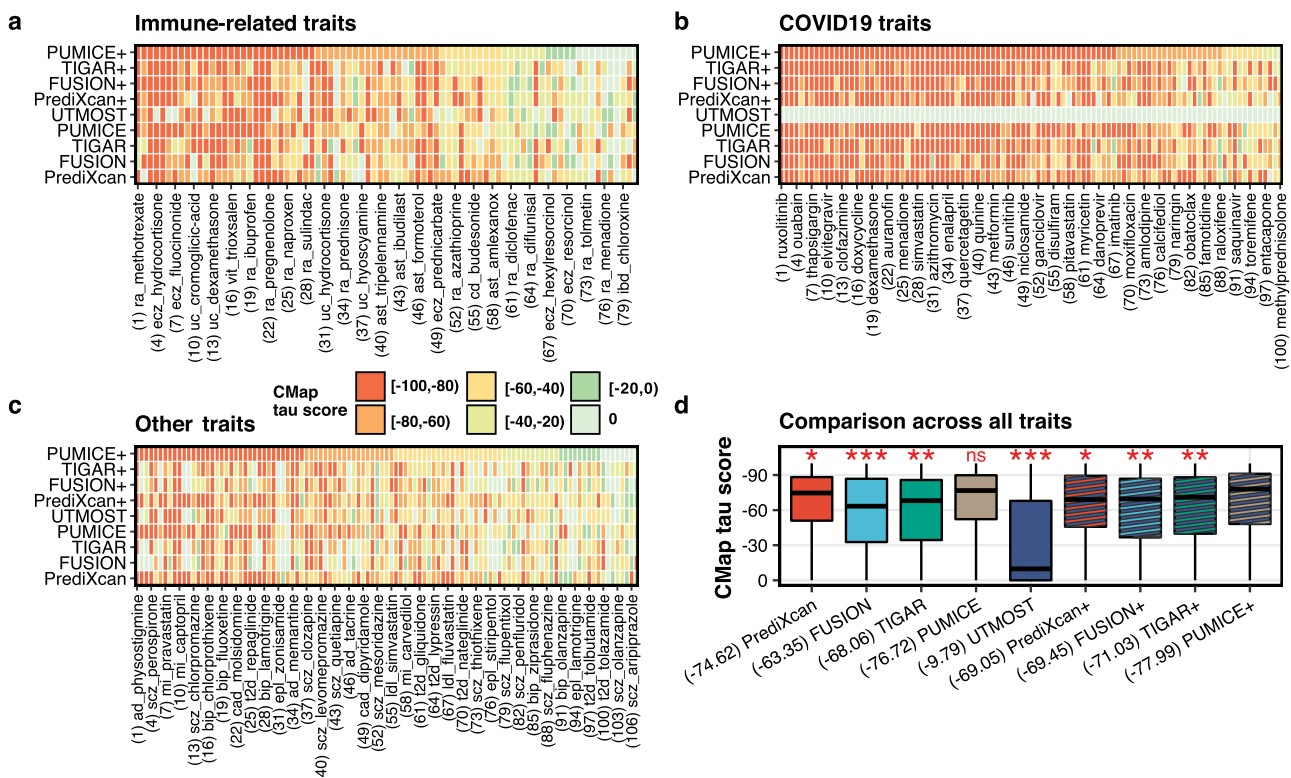

**Fig. 6 Computational drug repurposing predictions for drugs with known indications in 23 traits.** Panels (**a–c**) illustrates the heatmap of CMap scores derived from different TWAS methods for (**a**) immune-related traits, (**b**) COVID-19 traits, and (**c**) other traits. Due to the large number of trait-drug pairs, we only display text description of one for every three trait-drug pairs in the plot. Numbers in the parentheses are the indices of the displayed trait-drug pairs in the full list (Supplementary Data 16). Panel (**d**) displays the distribution of CMap scores across 23 traits. Median value is denoted in the parentheses. PUMICE+ achieves the most negative median value of CMap score (−77.99), which shows that it identifies putative target genes that are most consistent with target genes of approved drugs. P-values are based on the comparison between PUMICE+ and other methods using one-sided median test. The label "ns" denotes not significant; * denotes significance at p < 0.05; ** denotes significance at p < 0.01; *** denotes significance at p < 0.001. One-sided P-values are provided in the source data file. Minima and maxima values (excluding outliers) are represented by the lower- and upper-bound of the whiskers. Median value is represented by the bold line in the middle. First and third quartiles are represented by the lower- and upper-bound of the box. Trait abbreviations: ad = Alzheimer's disease; ast = asthma; bip = bipolar disorder; cad = coronary artery disease; cd = Crohn's disease; ecz = atopic dermatitis; epl = epilepsy; ibd = inflammatory bowel disease; ldl = low-density lipoprotein level; mi = myocardial infarction; ra = rheumatoid arthritis; scz = schizophrenia; t2d = type 2 diabetes; uc = ulcerative colitis; vit = vitiligo.

subacute post-COVID parkinsonism syndrome[49]. Importantly, one study suggested that COVID may act as an environmental trigger in the development of Parkinson's disease in genetically predisposed asymptomatic carriers (e.g., heterozygous variant in the *GBA* gene)[50]. *TRIM23* is critical for autophagy-mediated restriction of multiple viruses[51]. ATP11A functions as phospholipid flippase and is essential for enabling precursor B cells to flee engulfment by macrophages[52]. *TYK2* encodes tyrosine kinase 2 and is one of four gene targets for JAK inhibitors such as baracitinib[53].

For GWAS of hospitalized COVID vs. not hospitalized COVID (COVID-B1 dataset), we identified a novel *PXT1* gene (Fig. 7b). Interestingly, *PXT1* encodes a male germ cell-specific peroxisomal protein that could induce germ cell apoptosis and leads to infertility in male mice[54]. This finding may help explain the finding of the subsequent damage of male reproductive system by the COVID-19 infection[55].

For GWAS of hospitalized COVID vs. population (COVID-B2 dataset), we identified various biologically-relevant genes, including *RAD54L*, *MDC1*, *THBS3*, *MYO1C*, *DPP9*, and *NAPSA* (Fig. 7c). Both RAD54L and MDC1 are involved in DNA repair, and it was previously found that the activation of DNA damage response is one of the mechanisms exploited by coronavirus to allow viral replication and progeny production[56]. *THBS3* was

previously shown to decrease lung function[57]. MYO1C is abundantly expressed in B cells, especially in microvilli, and may have a role in immunologic synapse formation and antigen presentation[58]. *DPP9* encodes a serine protease that serves key roles in cleaving antiviral signaling mediator, antigen presentation, and inflammasome activation[59–61]. NAPSA, a secreted protein expressed in type II pneumocytes, is associated with lung surfactant processing[62].

For GWAS of COVID vs. population (COVID-C2 dataset), we identified *RBBP5*, *TFR2*, *GSDMA*, and *TULP2* (Fig. 7d). Circulating RBBP5 was independently identified as positively correlated with C-reactive protein, an established marker of poor prognosis in COVID[63]. *TFR2* encodes transferrin receptor 2 and is involved in iron transport. Dysregulation of iron homeostasis may play a key role in pathogenesis of viral infections[64]. GSDMA was recently identified as the mediators of pyroptosis, an inflammatory cell death triggered by cytosolic sensing of invasive infection. Importantly, GSDMA can form cell membrane pores upon activation and releases inflammatory cytokines into the extracellular space[65]. TULP2 was identified as a new RNA-binding protein required for mouse spermatid differentiation and male fertility[66].

Lastly, we performed CDR predictions on three COVID-19 datasets (COVID-A2, COVID-B2, and COVID-C2). By grouping

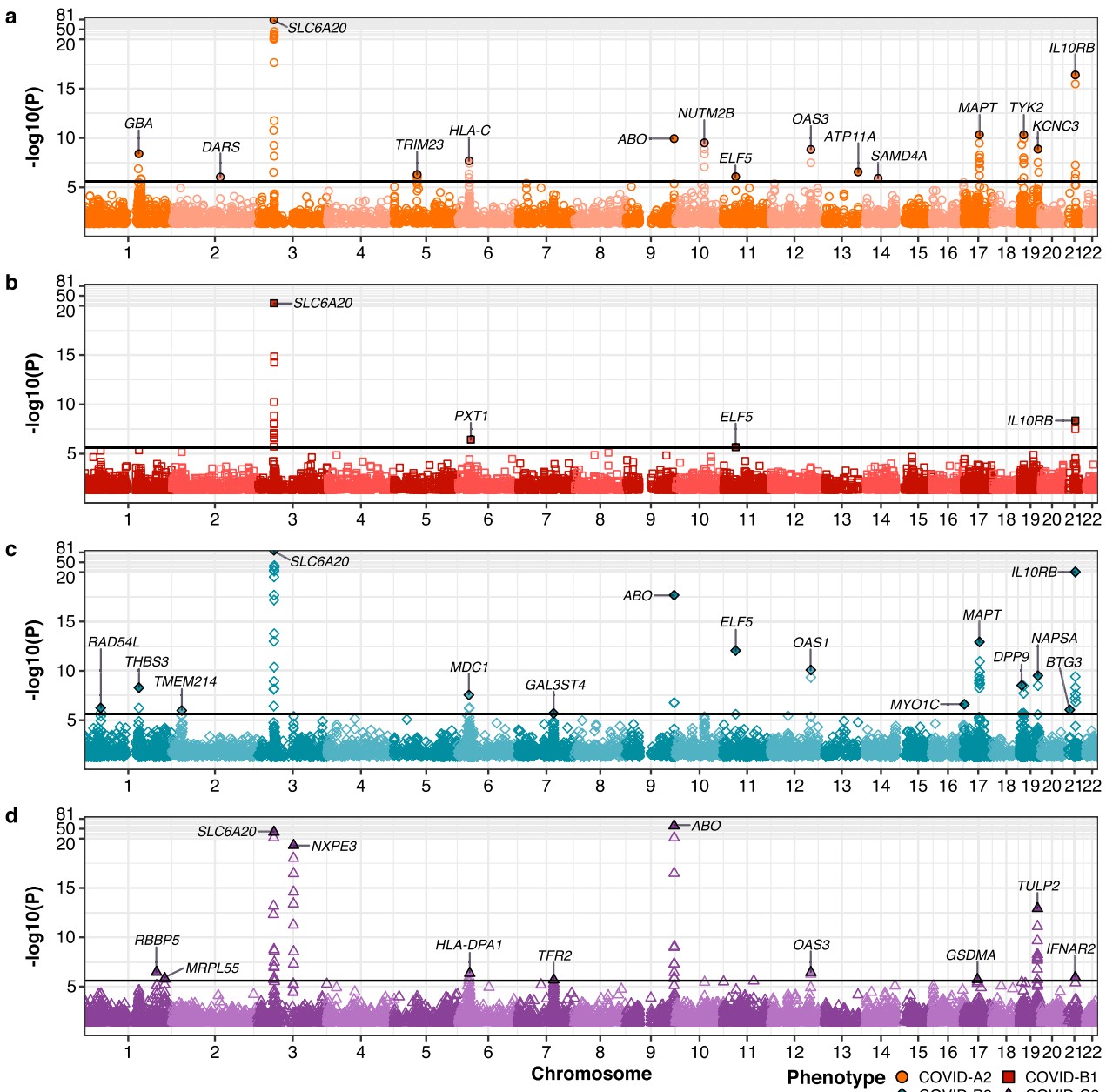

**Fig. 7 TWAS Manhattan plot for COVID-19-related outcomes via PUMICE+. a–d** illustrates the Manhattan plot for (**a**) COVID-A2, (**b**) COVID-B1, (**c**) COVID-B2, and (**d**) COVID-C2. Black horizontal line marks the genome-wide significance threshold at $2.5 \times 10^{-6}$ (Bonferroni threshold corrected for 20,000 genes). The most significant genes at each phenotype-locus pair are labelled. Two-sided $P$ value associated with each gene is calculated according to the TWAS Z-score for gene-based association test.

drugs according to perturbational classes and focusing on drug classes that appear in multiple sets of CDR predictions, PUMICE+ frequently identified protein synthesis inhibitor, proteasome inhibitor, bromodomain inhibitor, mTOR inhibitor, leucine-rich repeat kinase inhibitor, and *JAK* inhibitor as top drug classes identified. Importantly, emetine, a protein synthesis inhibitor, was shown to have the antiviral effect against SARS-CoV-2[67]. Previously, a proteasome inhibitor was shown to have an anti-inflammatory effect and an ability to inhibit coronavirus replication[68,69]. Therefore, proteasome inhibitor may be able to counteract both virus replication and cytokine storm[70]. The protein interaction map further reveals the synergy between SARS-CoV-2 and bromodomain proteins and bromodomain inhibitor might disrupt this interaction[71]. Immunoregulation with mTOR inhibitor or leucine-rich repeat kinase inhibitor

might also alleviate COVID-19 severity by halting cytokine storm[72–75]. JAK inhibitor is independently predicted to have anti-viral and anti-cytokine activity[76]. Baracitinib, a JAK inhibitor, is currently being evaluated in the Adaptive COVID-19 Treatment Trial for its safety and efficacy when used in combination with antiviral drug remdesivir.

## Discussion
In this work, we present a method that integrates genomic, transcriptomic, epigenomic, and 3D genomic data to identify target genes associated with complex traits/diseases. By integrating functionally relevant epigenomic and 3D genomic data, we observed significant improvement of gene expression prediction performance compared to other previous TWAS methods. We attribute this improvement to the integration of prior knowledge,

which guided the selection of biologically informative models. Improved transcriptome imputation performance leads to increased power to detect more trait-associated genes and improves downstream fine-mapping and drug repurposing analysis.

There have been some interests in the comparison between single-tissue and multi-tissue prediction models. Multi-tissue methods are useful, when the sample sizes from certain tissues are small, as they can borrow strength from other tissues. Yet, multi-tissue methods need datasets like GTEx where gene expression from multiple tissues are measured on the same individual. Multi-tissue methods also tend to mitigate the differences between tissues. As we showed in Results, when applied to TWAS, single-tissue prediction models allow the identification of many more tissue-specific gene x trait associations. Importantly, as functional genomic data is being rapidly generated, training sample sizes for most tissues will likely grow. Single-tissue methods tend to catch up and outperform multi-tissue methods when training sample sizes get bigger. Therefore, we foresee a more important role of single-tissue methods in the future.

TWAS may be complemented by other integrative approaches to study complex traits. For example, the work by Wu et al. seeks to combine single variant association tests based on enhancer-promoter interactions information[77], which may identify additional gene x trait associations. PUMICE, as a TWAS method, can naturally be used together with these complementary methods.

PUMICE framework showcased the power of an integrative method in gene discovery, fine-mapping, and clinical translation. Several future directions in data generation and method development could further enhance PUMICE and future methods thereof.

First, PUMICE would benefit from higher resolution 3D genomic data from matched tissues and cell types. Tissue/cell-type-specific 3D genomic data are still currently limited; hence, we used 3D genomic data from proxy tissues in the PUMICE model. It was based upon the assumption that tissues with similar global gene expression profiles have similar 3D genome organizational structures. This assumption was supported by the observation that domains are stable across different cell types[13] and 3D genome structures are correlated with gene expression profiles[78]. Nonetheless, we believe that 3D genomic data will become more widely available in the future as Hi-C method requires less starting material to perform experiments[79]. In addition, we also envision that integrating richer 3D genomic and epigenomic data from matched tissues would also help models that jointly consider cis- and trans-eQTLs.

Second, it is possible in the future to incorporate transcription factors (TFs) into PUMICE model. Enhancers and their associated TFs have a leading role in the initiation of gene expression[80]. While our method incorporates the information of enhancer sites, enhancers merely act as operational platforms to recruit TFs[80]. Therefore, further improvement can be expected from incorporating TF chromatin immunoprecipitation followed by sequencing data or TF binding models[81].

Third, PUMICE framework would benefit from samples of non-European ancestry for fine-mapping causal genes[82]. Through integrative approaches, PUMICE+ offers the narrowest credible sets among all methods. Yet, complex LD patterns can still be a major hurdle to narrow down the credible sets to single-gene resolution. As LD patterns vary across ancestries, integrating non-European samples and developing relevant methods would allow further refinement of causal genes, and facilitate downstream functional follow-up.

Lastly, current TWAS methods model gene expression levels with linear models, ignoring the possibility of epistasis where multiple regulatory variants jointly affect gene expression (e.g.,

enhancer-promoter interactions). Hence, it is of interest in the future to model gene expression levels with machine-learning methods that can handle non-linear relationships (e.g., random forest, boosting) as well as incorporating functional genomic data.

In conclusion, PUMICE is an integrative and flexible framework to perform gene-based association analysis. PUMICE is built on the wealth of multi-omics data that are publicly available, allowing the framework to identify prediction models with higher imputation accuracy. We envision that the PUMICE software pipeline and pre-computed models can benefit numerous upcoming in-silico functional follow-up studies and help retrieve useful information from the wealth of GWAS data.

## Methods

**GTEx genomic and expression data**. We obtained genotype data for 635 human donors from GTEx project (version V7, dbGAP accession code phs000424). Out of 635 donors, we identified 527 donors to be of European ancestry via ADMIXTURE[83] using 1000 Genomes project Phase3 data[84] as a reference panel (Supplementary Notes and Supplementary Data 18). We only focused on samples with >90% European ancestry composition.

We obtained normalized gene expression data directly from dbGAP, where we identified 513 human donors with European ancestry across 48 tissues with normalized gene expression levels. There is slight discrepancy between the number of donors of DNA sequence and expression data, as not all donors donated the same tissue types. The sample sizes of different tissues are shown in Supplementary Data 18, ranging from 69 (brain substantia nigra) to 405 (skeletal muscle). In our analysis, we adjusted gene expression data with the following covariates: 1) sex, 2) sequencing platform, 3) top 3 genetic principal components, and 4) probabilistic estimation of expression residuals (PEER) factors. As previously recommended, we applied 15 PEER factors for tissues with sample sizes ≤ 150, 30 PEER factors for tissues with sample sizes between 150 and 250 and 35 factors for tissues with sample sizes > 250.

**3D genomic and epigenomic data**. We obtained 3D genomic data from 3dgenome.org (for loop and TAD in 15 tissues)[85], GSE63525 (for domain in 4 tissues)[86], and GSE86189 (for pcHi-C in 18 tissues)[87]. All 3D genomic data are in BED format and the list of available tissues and primary cell lines are listed in Supplementary Data 19. Given that 3D genomic data are not available in every GTEx tissue, we use 3D genomic data from tissues with a similar global expression profiles as proxy, as 3D genome organizational structure is closely correlated with gene expression profiles[78] and remain stable among similar tissues/cell types[13] (Supplementary Data 20). In order to determine the similarity among tissues, we used CountClust to cluster samples of RNA sequencing data using the grade of membership (GoM) models[88]. Specifically, FitGoM function in CountClust package was applied to GTEx Analysis V7 gene read counts file with the number of clusters being K = 20 (for 48 tissues) and K = 6 (for 13 brain tissues) as recommended. It should be noted that we filtered out samples derived from non-European human donors (resulting in a total of 8,547 samples across 48 tissues and 1,314 samples across 13 brain tissues). The structure and heatmap plots are shown in Supplementary Figs. 10-12.

We also obtained epigenomic annotation data from Encyclopedia of DNA Elements (ENCODE) for 48 tissues[89,90]. Specifically, we used a list of candidate epigenomic marks, including H3K27ac, H3K4me3, DNase hypersensitive, and CTCF. All epigenomic annotation data are also in BED format and the list of available tissues and primary cell lines and corresponding ENCODE accession IDs are given in Supplementary Data 21. BEDTools was used to identify SNPs overlapping with epigenomic tracks[91].

**PUMICE model description**. The PUMICE model separately considers essential and non-essential predictors depending on whether they overlap with epigenomic annotation tracks. It seeks to minimize the following loss function

$$L(\boldsymbol{\beta_1}, \boldsymbol{\beta_2}; \lambda, \phi) = \|\mathbf{E} - \mathbf{X_1}\boldsymbol{\beta_1} - \mathbf{X_2}\boldsymbol{\beta_2}\|_2^2 + \frac{1}{2} \times \frac{\lambda}{2}(\phi\|\boldsymbol{\beta_1}\|_2^2 + \|\boldsymbol{\beta_2}\|_2^2)$$
$$+ \frac{1}{2}\lambda(\phi\|\boldsymbol{\beta_1}\|_1^1 + \|\boldsymbol{\beta_2}\|_1^1)$$

(1)

where $\mathbf{X_1}$ is the essential predictor overlapping functional annotations, $\mathbf{X_2}$ is the nonessential predictors, and $\boldsymbol{\beta_1}, \boldsymbol{\beta_2}$ are weights of essential and non-essential variants used to predict gene expression levels. The numbers of essential and non-essential predictors are denoted by $p_e$ and $p_{ne}$. The model takes advantage of both LASSO and ridge penalties. Collectively, we call the weight parameters as $\boldsymbol{\beta} = (\boldsymbol{\beta_1}, \boldsymbol{\beta_2})$.

We develop a cyclic coordinate descent algorithm to fit the model. We iteratively update the parameters $\boldsymbol{\beta_1} = \left(\beta_{1,1}, \dots, \beta_{1P_e}\right)$ and $\boldsymbol{\beta_2} = \left(\beta_{2,1}, \dots, \beta_{2p_{ne}}\right)$, one at a time until the model converges.

Specifically, we set the initial parameter values $\hat{\boldsymbol{\beta}}_1^{(0)}$, $\hat{\boldsymbol{\beta}}_2^{(0)}$ to be the least square estimates. At iteration $a$, we update the coefficient for the $j^{th}$ predictor $\beta_{1j}$ as

$$
\hat{\beta}_{1j}^{(a)} = \begin{cases} (\frac{1}{N}\mathbf{X}_{1j}^{\mathbf{T}}\mathbf{r}_{1j}^{(\mathbf{a})} - \alpha\lambda\phi)/(\frac{1}{N}\mathbf{X}_{1j}^{\mathbf{T}}\mathbf{X}_{1j} + (1-\alpha)\lambda) & \text{if } \frac{1}{N}\mathbf{X}_{1j}^{\mathbf{T}}\mathbf{r}_{1j}^{(\mathbf{a})} > \alpha\lambda\phi \\ 0 & 0 \le \frac{1}{N}\mathbf{X}_{1j}^{\mathbf{T}}\mathbf{r}_{1j}^{(\mathbf{a})} \le \alpha\lambda\phi \\ (\frac{1}{N}\mathbf{X}_{1j}^{\mathbf{T}}\mathbf{r}_{1j}^{(\mathbf{a})} + \alpha\lambda\phi)/(\frac{1}{N}\mathbf{X}_{1j}^{\mathbf{T}}\mathbf{X}_{1j} + (1-\alpha)\lambda) & \frac{1}{N}\mathbf{X}_{1j}^{\mathbf{T}}\mathbf{r}_{1j}^{(\mathbf{a})} < -\alpha\lambda\phi \end{cases} \quad (2)
$$

where $\hat{\beta}_{1j}^{(a)}$ is the updated parameter value at iteration $a$, $r_{1j}^{(a)}$ is the residual as defined by $\mathbf{r}_{1j}^{(\mathbf{a})} = \mathbf{E} - \mathbf{X}_{1,-j}\hat{\boldsymbol{\beta}}_{1,-j}^{(\mathbf{a-1})} - \mathbf{X}_2\hat{\boldsymbol{\beta}}_2^{(\mathbf{a-1})}$, and $\hat{\beta}_{1,-j}^{(a-1)} = \left(\hat{\beta}_{1,1}^{(a-1)}, \ldots, \hat{\beta}_{1,j-1}^{(a-1)}, \hat{\beta}_{1,j+1}^{(a-1)}, \ldots \hat{\beta}_{1,p_e}^{(a-1)}\right)$. Similarly, we update the coefficients for non-essential predictors using the following:

$$
\hat{\beta}_{2j}^{(a)} = \begin{cases} (\frac{1}{N}\boldsymbol{X}_{2j}^T r_{2j}^{(a)} - \alpha\lambda)/(\frac{1}{N}\boldsymbol{X}_{2j}^T\boldsymbol{X}_{2j} + (1-\alpha)\lambda) & \text{if } \frac{1}{N}\boldsymbol{X}_{2j}^T r_{2j}^{(a)} > \alpha\lambda \\ 0 & 0 \le \frac{1}{N}\boldsymbol{X}_{2j}^T r_{2j}^{(a)} \le \alpha\lambda \\ (\frac{1}{N}\boldsymbol{X}_{2j}^T r_{2j}^{(a)} + \alpha\lambda)/(\frac{1}{N}\boldsymbol{X}_{2j}^T\boldsymbol{X}_{2j} + (1-\alpha)\lambda) & \frac{1}{N}\boldsymbol{X}_{2j}^T r_{2j}^{(a)} < -\alpha\lambda \end{cases} \quad (3)
$$

**Simulation Study Design**. We performed extensive simulation studies to evaluate the performance of PUMICE and other TWAS methods. Briefly, we simulated gene expression values based on real genotypes from two datasets: 1) GTEx (as training and validation data) and 2) CMC (for as external test data). We then simulated GWAS Z-scores following the framework of Feng et al.[92]

To evaluate the single-tissue TWAS method, we modified the simulation framework from Nagpal et al.[8]. Gene expressions for gene $g$ are simulated based on the linear model:

$$
\mathbf{E_g} = \mathbf{X_g}\left[\mathbf{w_{epi}}, \mathbf{w_{notepi}}\right] + \boldsymbol{\varepsilon_e} \quad (4)
$$

where $\mathbf{X_g}$ is the matrix of normalized genotypes (with mean of 0 and variance of 1) of eQTL variants for gene $g$, with columns arranged in the order of the $M_{epi}$ essential variants overlapping epigenomic annotation and the $M_{notepi}$ non-essential variants that do not overlap annotations. The corresponding weights are given by $\mathbf{w_{epi}}$ and $\mathbf{w_{notepi}}$. We assume that the gene expression variance explained by cis-regulatory variants is $h_e^2$. The weights are simulated according to normal distributions $\mathbf{w_{epi}} \sim MVN(0, h_{epi}^2/M_{epi}\mathbf{I_{M_{epi}}})$ and $\mathbf{w_{notepi}} \sim MVN(0, h_{notepi}^2/M_{notepi}\mathbf{I_{M_{notepi}}})$, where $h_{epi}^2$ and $h_{notepi}^2$ are expression variance explained by essential and non-essential variants. To characterize the possible enrichment of heritability for essential variants, we define an enrichment factor (EF) as

$$
EF = \left(h_{epi}^2/M_{epi}\right)/(h_{notepi}^2/M_{notepi}) \quad (5)
$$

The residual error is assumed to follow $\boldsymbol{\varepsilon_e} \sim MVN\left(0, \left(1-h_e^2\right)\mathbf{I_{N_{train}}}\right)$, where $h_e^2$ is the expression variance explained by cis-regulatory variants, and $N_{train}$ is the training sample sizes. Finally, we simulate GWAS Z-scores by randomly sampling from multivariate normal distribution $\mathbf{Z} \sim MVN(\Sigma_\mathbf{G} \times \sqrt{N_{GWAS} \times h_p^2} \times \left[\mathbf{w_{epi}}, \mathbf{w_{notepi}}\right], \Sigma_\mathbf{G})$ where $\Sigma_\mathbf{G}$ is the LD structure of the cis-SNPs (calculated from 5,000 randomly selected samples of European ancestry from UK Biobank), $N_{GWAS}$ is the GWAS sample size (set to 500 K), and $h_p^2$ is the phenotypic variance explained by the gene expression.

We considered a number of scenarios by varying

1) The number of causal SNPs ($n_{causal} \in \{2, 8, 32, 64\}$);
2) The proportion of causal SNPs in the epigenomic regions ($p_{causal_{epi}} \in \{0.4, 0.8, 1\}$),
3) The window containing causal SNPs ($w \in \{\pm 1Mb, \pm 250kb, 3D \text{ window}, \text{mixed}\}$) where mixed refers to the simulation that randomly selects one window from $\pm 1Mb$, $\pm 250kb$, and 3D windows for each gene.
4) The expression and phenotypic heritability pairs ($(h_e^2, h_p^2) \in \{(0.025, 0.02), (0.05, 0.01), (0.1, 0.005)\}$), and
5) Epigenomic enrichment factor ($EF \in \{2, 4\}$).
6) The training sample sizes from 100, 200, and 300.

To compare with multi-tissue method, we modified the simulation framework of Feng et al.[92] to allow the incorporation of epigenomic and 3D genomic annotations. In the framework, we designated a causal tissue where the genetic variants directly influence expression levels. Due to genetic correlations between gene expression levels of causal and other tissues, genetic variants may influence the gene expression levels of non-causal tissues as well but with weaker effects.

To model gene expression levels, we utilized real genotype data from GTEx, which comprised of 513 donors across 48 GTEx tissues, and made sure to retain sample sizes of different tissues as the original GTEx data. We specifically assigned Brain_Frontal_Cortex_BA9 tissue as a causal tissue. Gene expression levels of the causal tissue are simulated in the same manner as single tissue analysis, i.e.,

$$
\mathbf{E_{g_{causal}}} = \mathbf{X_g}\left[\mathbf{w_{causal\,epi}}, \mathbf{w_{causal\,notepi}}\right] + \boldsymbol{\varepsilon_e} \quad (6)
$$

We randomly select $N_{corr}$ correlated tissues, whose eQTL effects are correlated with the causal tissue. We assume the genetic correlation $\rho$ between the causal tissue and correlated tissues, and generated the effects by

$$
\mathbf{w_{corr}} = MVN\left(\rho\left[\mathbf{w_{causal_{epi}}}, \mathbf{w_{causal_{notepi}}}\right], (1-\rho^2) \times h_e^2 \times \mathbf{I_{N_{corr}}}\right) \quad (7)
$$

Residual gene expression values of correlated tissues were simulated as $\boldsymbol{\varepsilon_{e'}} = MVN(0, \mathbf{diag}(\sqrt{1-h_e^2}) \times \Sigma \times \mathbf{diag}(\sqrt{1-h_e^2}))$, where $\Sigma$ is the residual correlation among gene expression levels across tissue. Here we set $\Sigma = \mathbf{diag}(1)$.

Gene expression levels of correlated tissues are simulated by

$$
\mathbf{E_{g_{corr}}} = \mathbf{X_g}\mathbf{w_{corr}} + \boldsymbol{\varepsilon_{e'}} \quad (8)
$$

We also designate uncorrelated tissues, and simulate the gene expression using the same model as the correlated tissues but set $\rho = 0$.

We simulated GWAS Z-scores by randomly sampling from the multivariate normal distribution $\mathbf{Z} \sim MVN(\Sigma_\mathbf{G} \times \sqrt{N_{GWAS} \times h_p^2} \times \left[\mathbf{w_{causal\_epi}}, \mathbf{w_{causal\_notepi}}\right], \Sigma_\mathbf{G})$. Due to the large computational cost of UTMOST and the huge number of combinations of parameters, we repeated simulations for 1000 times for each of the five pairs of $(\rho, N_{corr})$ values, i.e., (0.3, 0), (0.3, 24), (0.3, 47), (0.7, 24), and (0.7, 47). For each replicate, we randomly sample $n_{causal}$, $p_{causal\_epi}$, $w$, $h_e^2$, $h_p^2$, and $EF$ parameters. Results are reported for each $(\rho, N_{corr})$ value based upon the averages across replicates.

Under the null hypothesis ($h_p^2 = 0$), TWAS type I errors were calculated as the fraction of heritable genes showing significant $p$ values among all simulated genes ($p < 0.05$). Under the alternative hypothesis, power was calculated as the fraction of heritable genes with significant TWAS $p$ values among all simulated genes ($p < 5 \times 10^{-5}$, the Bonferroni threshold for 1000 simulated genes).

**Model Assessment in GTEx Using Cross-Validation**. We randomly split the data into five folds. For each iteration, we set 4/5 of the data as training fold and the remaining 1/5 as test fold. For methods that require parameter tuning (i.e., PrediXcan, EpiXcan, PUMICE, and UTMOST), we performed nested cross validation, where we further performed 5-fold cross-validation within training data to select tuning parameters that minimize mean squared error of the model. We then re-trained the model in the training fold (4/5 of the entire dataset) using selected tuning parameters and evaluated the performance in the testing fold (1/5 of the entire dataset) via Spearman's rank correlation coefficient ($r$). For methods that do not require parameter tuning (i.e., FUSION and TIGAR), we trained the model directly using training folds and evaluated the performance in the testing fold. We repeated this process five times. The final reported correlation coefficient was calculated based on the average of the five Spearman correlation estimates. To estimate the statistical significance of the prediction models, we followed PrediXcan pipeline which uses Stouffer's method to convert averaged correlation to z-score and then to $p$-value.

A model was deemed significant if the average of Spearman's correlation coefficients was greater than 0.1 and the estimated $p$-value for this statistic was less than 0.05, as suggested in earlier work[93].

**Gene x trait association analysis**. For each tissue, we imputed the expression of significant models into GWAS summary-level statistics using Gaussian imputation[94]. We then calculated the TWAS statistic using the following formula

$$
S = \frac{\boldsymbol{\beta}^\mathbf{T}\mathbf{Z}}{(\boldsymbol{\beta}^\mathbf{T}\Sigma\boldsymbol{\beta})^{\frac{1}{2}}} \quad (9)
$$

where $\boldsymbol{\beta}$ is the weight vector from gene expression prediction model, $\mathbf{Z}$ is a vector of GWAS Z-scores, and $\Sigma$ is the regularized LD covariance matrix (to ensure the matrix is invertible) among SNPs within the model. A positive TWAS statistic indicates that genetically regulated overexpression of a gene is associated with increased disease risk (or trait values), while a negative z-score indicates that genetically regulated overexpression of gene is associated with decreased disease risk (or trait values). Gene x trait association is considered significant if the $p$-value is less than the Bonferroni threshold (0.05/the number of tests performed for each tissue). The $p$-value threshold used for each tissue can be found in Supplementary Data 22.

**Cauchy combination test in PUMICE+**. PUMICE leverages both 3D genomic and epigenomic data, which improves performance over other single-tissue TWAS methods. On the other hand, shared eQTLs across tissues, if properly utilized, will further improve the power for TWAS. Importantly, single-tissue and multi-tissue prediction models complement each other, as single-tissue prediction models are better at identifying unique gene x trait associations. We sought to combine each single-tissue method with the multi-tissue method UTMOST in our analysis. Numerous methods exist to combine $p$-values from correlated test statistics. We

chose to implement the combined method using Cauchy combination due to its good power and its simplicity[95]. In our analysis, we calculated the $p$ values using Cauchy combination test to combine each single tissue method with UTMOST $p$-values using equal weights.

**Fine-mapping TWAS results**. It is of interest to narrow down the list of causal genes. We conducted standard TWAS fine-mapping using FOCUS for each TWAS method that is not based on combined $p$-values. Specifically, we created a weight database specific for each tissue and run FOCUS for each tissue separately. We then compared the size of the credible sets and established the advantage of PUMICE over other methods that are not based on combined $p$-values.

The PUMICE+ and other "+" methods are based upon combined $p$-values, for which FOCUS cannot be applied. We performed fine-mapping of TWAS hits by extending existing methods[38,96] based upon Gaussian copula. We first transformed two-sided TWAS $p$-value of each gene to Z-score statistic [i.e., $Z = 1 - \Phi^{-1}(p)$, where $\Phi$ is the cumulative distribution function for standard normal random variables], then estimated the correlation between converted Z-scores, and finally calculated Bayes factors and posterior inclusion probabilities to quantify the probability that each gene was causal. This follows the same idea as other GWAS/TWAS fine-mapping methods using approximate Bayes factors[96,97], except that we convert Z-scores to $p$-values. However, using copula, this method allows us to work with a broader class of statistical methods including the ones based upon combined $p$-values. Our method first assumes that each locus contains a single causal gene. For locus that contains multiple causal genes, we follow existing work, perform conditional analysis by conditioning on the top gene, and conduct fine mapping for secondary signals using the conditional analysis results.

Based upon the workflow in FOCUS, for each tissue, we defined the loci iteratively. We first defined the top locus as a LD block (based on LDetect[98]) surrounding the top gene-level association. For remaining genes outside the top locus, we define the second locus surrounding the most significant gene. We repeated the procedure to define additional loci until we exhausted the list of significant genes.

The gene-level association $p$-values are denoted by $\widetilde{p}_1, \ldots, \widetilde{p}_g$ and the single SNP $p$-values in the gene region that are not used in gene expression models are denoted as $p_1, \ldots, p_v$. We converted these $p$-values to Z-score statistics using inverse normal transformation, which we denoted as $\widetilde{T}_1, \ldots, \widetilde{T}_g, T_1, \ldots, T_v$ (i.e. $T = \Phi^{-1}(1 - p)$). Next, we estimated correlations between converted statistics using bootstrap. Under the null hypothesis, the single variant association statistics follow multivariate normal distribution, with correlation matrix equal to the LD coefficients. To calculate the correlation between statistics $\widetilde{T}_1, \ldots, \widetilde{T}_g, T_1, \ldots, T_v$, we employed a Monte Carlo approach. For any two genes, e.g., gene 1 and 2, we denoted the single variant association statistics of the SNPs used in the gene expression model as $Z_{1,1}, \ldots, Z_{1,m_1}$ and $Z_{2,1}, \ldots, Z_{2,m_2}$, and their covariance matrix as $\Sigma_{Z_1, Z_2}$. To calculate covariance between $\widetilde{T}_1$ and $\widetilde{T}_2$, we repeated the following steps 1000 times.

At iteration $l$:

- We simulated $Z_{1,1}^{(l)}, \ldots, Z_{1,m_1}^{(l)}$ and $Z_{2,1}^{(l)}, \ldots, Z_{2,m_2}^{(l)}$ according to multivariate normal distribution $MVN(0, \Sigma_{Z_1, Z_2})$.
- Based upon simulated Z-scores, we calculated PUMICE+ statistic and $p$ values.
- We converted PUMICE+ $p$ values to Z-score statistics, which we denoted as $\widetilde{T}_1^{(l)}$ and $\widetilde{T}_2^{(l)}$.

The correlation between gene-level statistics $\widetilde{T}_1$ and $\widetilde{T}_2$ are estimated using the empirical correlation between $\widetilde{T}_1^{(l)}, \widetilde{T}_2^{(l)}$.

To estimate the posterior inclusion probability, we assumed one causal gene per locus, and performed iterative conditional analysis to identify secondary associations and used the conditional $p$ value to fine map secondary genes. We calculated the approximate Bayes factor for gene $g_0$ using the approximate Bayes factor:

$$ABF(g_0) = \frac{1}{\sqrt{\psi^2 + 1}} \exp\left(\frac{\psi^2 Z_{g_0}^2}{2(1 + \psi^2)}\right) \tag{10}$$

where $\psi$ is the prior assigned to the effect sizes and we set it to 0.05 as suggested in Wakefield et al[96].

The PIP for gene $g_0$ can be calculated by

$$PIP(g_0) = \frac{ABF(g_0)}{\sum_l ABF(l)} \tag{11}$$

When multiple association signals are present, conditional analysis is performed conditioning on the top gene/variants in each locus. We will perform conditional analysis of the Z-scores $\widetilde{T}_j$ or $T_j$ over the Z-score of the top signal and obtain the residual. It is well known that the conditional analysis can be performed using summary-level Z-scores and the covariance matrix between Z-scores (as estimated using the Monte Carlo method) without individual level data[99]. The conditional Z-scores will be used in the fine mapping of secondary signals.

**Computational drug repurposing**. For each trait, we first identified genes with significant $p$-values (below the Bonferroni threshold of testing multiple genes) in each tissue. For genes that are associated with a trait in multiple tissues, the associations were summarized as an averaged Z-score. Only traits with known drug indications and having at least 10 positively and 10 negatively associated genes (in at least 4 out of 9 TWAS methods) were investigated as recommended[23,24]. This resulted in a total of 23 traits for subsequent drug repurposing analysis (Supplementary Data 15).

We compared TWAS signatures to the gene expression patterns caused by perturbagens in the CMap database (from L1000 assay). Here, we only focused on the touchstone dataset of CMap which encompassed gene expression patterns across nine cell lines treated with ~3000 well-annotated small molecule drugs. In order to link disease to drug-induced state, CMap calculates a $\tau$-score to assess the correlation between a query signature (a Z-score measuring how gene expression is associated with disease status) and a reference profile (measuring how a given component modifies gene expression). A negative $\tau$-score indicates that the identified molecule will normalize trait-associated gene expression profile, which can be repurposed to treat the disease[25].

In order to compare the accuracy of computational drug repurposing predictions across all TWAS methods, we compared drug repurposing predictions to known physician-curated indications. Drug indications are obtained from the CLUE Drug Repurposing Hub. For COVID traits, we downloaded proxy for drug indications from covid19-help.org and filtered for drugs that have at least 2 references. For each drug-disease pair, we calculated a minimum of $\tau$ values across nine touchstone cell lines of each TWAS method. A more negative $\tau$-score for a given TWAS method suggests that it identifies a larger number of putative targets for repurposing, as a larger fraction of its identified disease-associated genes can be reversed by approved drugs for treating the disease, and the target genes identified by the TWAS method are more consistent with approved drugs.

**Software URLs**. PUMICE software can be found at https://github.com/ckhunsr1/PUMICE[100]. PrediXcan software can be found at https://github.com/hakyimlab/MetaXcan. FUSION software can be found at https://github.com/gusevlab/fusion_twas. EpiXcan software can be found at https://bitbucket.org/roussoslab/epixcan/src/master/. TIGAR software can be found at https://github.com/yanglab-emory/TIGAR and CTIMP (UTMOST) software can be found at https://github.com/yiminghu/CTIMP.

**Reporting summary**. Further information on research design is available in the Nature Research Reporting Summary linked to this article.

## Data availability

The pre-trained GTEX v7 gene expression prediction models (hg19 reference genome) generated in this study can be found at https://github.com/ckhunsr1/PUMICE/tree/master/models[100]. GTEx V7 data can be obtained from dbGaP study accession phs000424.v7.p2. CMC data can be obtained from https://www.synapse.org with the following Synapse ID syn2759792. DGN data can be requested at https://www.nimhgenetics.org under "Depression Genes and Networks study (D. Levinson, PI)". GEUVADIS data can be accessed at https://www.ebi.ac.uk/arrayexpress/experiments/E-GEUV-1. The UKB analysis was conducted via application number 21037. The genotype data can be accessed through application at https://www.ukbiobank.ac.uk. Epigenomic data were obtained from http://screen.encodeproject.org. 3D genomic data were obtained from http://3dgenome.org. Computational drug repurposing analysis was conducted on CLUE Drug Repurposing Hub, which can be accessed at https://clue.io/repurposing-app. CLUE drug-disease indications can be obtained from https://clue.io/repurposing#download-data. COVID drug-disease indications can be obtained from https://covid19-help.org.

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

## Acknowledgements
This work was supported by the National Institutes of Health grants R01HG008983 (B.J., D.L.), R01GM126479 (B.J., D.L.), R56HG011035 (B.J., D.L.), R03OD032630 (B.J., D.L.), T32GM118294 (C.K.), T32LM012415 (C.K.). This work was also funded by the Lupus Research Alliance (L.C.) and CURE funds from the Pennsylvania Department of Health (L.C.). This work was also funded in part by generous support from Robert and Sevia Finkelstein (C.K.). We thank J. Yang for help with TIGAR method and implementation. We thank W. Zhang for providing us with EpiXcan data. We thank N. Mancuso for help with FOCUS method and implementation. This study utilizes summary statistics from many GWAS consortia. We thank the investigators in these GWAS consortia for generously sharing their data. We are also grateful for COVID-19 Host Genetic Initiative for immediate COVID-19 GWAS data release.

## Author contributions
C.K., L.C., B.J., and D.L. conceived the study and developed the statistical model. C.K. led the data analysis. C.K., D.M., R.S., F.C., L.Y., and L.W. conducted analyses. D.M., R.S., L.W., J.H., S.E., J.D.W., G.S., O.M. helped with data interpretation. C.K. and D.L. prepared the manuscript. All authors contributed to manuscript editing and approved the manuscript.

## Competing interests
The authors declare no competing interests.
