## [Peer Review File · Nature Communications]

Integrating 3D genomic and epigenomic data to enhance target gene discovery and drug repurposing in transcriptome-wide association studyREVIEWER COMMENTS

Reviewer #1 (Remarks to the Author):

The authors propose a very interesting TWAS method to account for 3D genomic and epigenomic data. The application study results show promising performance by the proposed new method. Here are my major concerns:

1. In the abstract, the authors write "Current approaches begin with a genomic interval of given size surrounding the gene start and end sites. This criterion lacks rigorous biological justification and hence results in less accurate models." I do not agree with this strong statement. Current approaches only consider eQTL within a specified window around gene start and end sites, which may show additional biological information can be used from additional 3D genomic and epigenomic data but not as the authors' statement of "lacks rigorous biological justification and results in less accurate models". As shown in the real application studies, the proposed method with a single tissue type won't outperform the multi-tissue TWAS method with a similar sample size. These results show the proposed method is comparable with some existing TWAS methods under certain situations.
2. Although the authors show "PUMICE resulted in an average of 30% and 16% improvement in imputation accuracy and an average gain of 52% and 12% more significant models" and "+ identified 51% more independent novel genes, increased mean chi-square statistics values at known loci by 26.5%, as well as achieved narrower credible interval for 73% of the traits being analyzed than the second-best method", the authors should also include simulation studies to compare gene expression prediction accuracy, TWAS power, and Type I error of the proposed method, with a comparison with existing methods.
3. It seems that the proposed method will do nested cross-validation to select the best window size for test SNPs. This is not clearly written in the main text. Since the test window size varies from gene to gene, tissue to tissue, comparing the credible interval size with the other TWAS methods that use 1MB region around gene start and end is not fair. Also, the discussion of the median number of test SNPs is also not fair.
4. How to define "independent" or "fine-mapping" significant gene is important and should be written clearly along with the motivation in the main text. How to define "significant" test gene should also be made clear in the main text.
5. Descriptions about these 3D genomics informed regions and epigenetic information would be helpful in the main text to help readers to understand why these would be biologically relevant.
6. The authors compared with many other TWAS methods besides PrediXcan and UTMOST, like FUSION and TIGAR. But only discussed the comparison results with respect to the number of significant test genes, median correlation values with PrediXcan and UTMOST. I think the authors should also include FUSION and TIGAR into the comparison.
7. Since even nested cross-validation prediction results would be affected by overfitted models, I think the authors should focus more on comparing the gene expression prediction accuracy with independent data sets instead of focusing on discussing the performance within GTEx tissues.
8. For gene expression prediction accuracy, comparing rank correlation might make more sense than person correlation.
9. In our studies, we noticed using the Z-score test statistic as used in this paper with non-standardized SNP weights derived from non-standardized reference gene expression traits and genotype data will cause huge inflation. If the SNP weights were derived from non-standardized

reference data in this paper, I would suggest using the S-PrediXcan test statistic for the gene-trait association study.

10. Text labels in the figures are too small to read.

Reviewer #2 (Remarks to the Author):

This manuscript describes a new method, PUMICE, to integrate genomic/epigenomic information for TWAS analysis. PUMICE first divides SNPs into two groups that include an essential group and a non-essential group. PUMICE then places different penalties on the SNP effects in two groups separately for building the gene expression prediction models. The authors show that PUMICE outperforms many TWAS methods in both internal and external predictions, and performs well in TWAS and TWAS fine-mapping analysis across multiple traits as well as for drug repurposing.

Overall, the paper is well written and easy to follow. The proposed method is also new. However, the benefits of PUMICE for TWAS analysis are not yet demonstrated in the paper, as the other methods are not compared in a fair way with PUMICE (main comment #4 below). In particular, PUMICE does not outperform the other TWAS methods for TWAS applications across the complex traits and only PUMICE+MASHR does. However, PUMICE+MASHR is a simple strategy that combines PUMICE with MASHR and can be equally applied to all other TWAS methods for them to be combined with MASHR. Therefore, a thorough and fair comparison of PUMICE+MASHR with other TWAS methods + MASHR would be critical for evaluating the performance of PUMICE in the TWAS applications. My main comments are listed below:

1. First, it is unclear how those non-penalization-based methods, including FUSION-BSLMM, TIGAR-VB, TIGAR-MCMC, were applied. Those methods can make use of the entire data, including both training and validation data, for model fitting. Consequently, those non-penalization-based methods can take advantage of the larger sample size in the combined training and validation data to improve prediction. It is important to fit those methods correctly and describe the fitting details in the methods section.

2. In the external prediction section, the authors concluded that "The performance of single-tissue TWAS methods improves with bigger training sample size and often surpasses multi-tissue methods in GTEx tissues with large sample sizes" (line 187-188). However, the benefits of using single-tissue with larger sample size do not appear to be explored in the later TWAS analysis section, where the GTEx data but not the other three single-tissue data was used for TWAS analysis on the 82 complex traits. It would be important to apply various TWAS methods on DGN, CMC, and GEUVADIS for the TWAS analysis of the 82 complex traits. Would PUMICE still work better than the other TWAS methods when single-tissue expression data is used for TWAS analysis of these complex traits?

3. PUMICE+MASHR requires the input of a p-value from MASHR. However, MASHR a Bayesian method and uses a local false sign rate (lfsr) to declare significance. How did you get a p-value from MASHR? Did you obtain a p-value from its Bayesian factor using Wakefield's method? If so, how do you justify the use of Wakefield's method, given that it only applies to a logistic regression model with one predictor variable?

4. PUMICE+MASHR is a simple strategy that combines the PUMICE p-value with the MASHR p-value. Such strategy can be easily applied to all other TWAS methods. To demonstrate the benefits of PUMICE, it would be important to compare PUMICE+MASHR with the performance of each other method + MASHR. For example, given that UTMOST yielded the lowest score for computational drug repurposing (CDR), would UTMOST+MASHR also work better than PUMICE+MASHR? Similarly, in Figure 4, DAPGW identified more unique genes than PUMICE and also fewer new genes, suggesting

potentially fewer false signals and more power than PUMICE; would DAPGW+MASHR also outperform PUMICE+MASHR?

5. For CDR, why examining only immune-related traits but not all the 82 traits? Does PUMICE perform better than the other methods for the other traits? I presume both positive and negative connectivity scores would be helpful for evaluating quantitative traits?

6. For the fine-mapping results, it would be helpful to compare the performance of PUMICE directly with the standard TWAS fine-mapping method FOCUS. FOCUS is the standard TWAS fine-mapping method and relies on a fine-mapping algorithm that is different from the one described in the method section.

Minor comments:

line 115: what is the parameter a ? Is this a typo? This parameter is not in the equation and not explained in the paragraph.

line 251: 2) -> 3)

line 258: 3) -> 4)

Reviewer #3 (Remarks to the Author):

The work presents a novel method for training gene expression prediction models for TWAS. The method uses penalized regression, with prior functional information on "essential" regions used to to specify a different penalty for variants in these regions than other variants. The method is rigorously evaluated against a variety of alternative TWAS approaches and yields higher accuracy in-sample and in independent data (I commend the authors for seeking out diverse independent datasets to evaluate performance). When used for TWAS, the method seems to identify a larger number of unique/independent associations than any other single-tissue approach, and does best when also combined with a multi-tissue approach. A COVID TWAS and drug repurposing study is carried out, but it is difficult to evaluate the significance of these findings since we have so little ground truth about the phenotype or effective treatments. Overall this is a very thorough analysis that investigates many aspects of TWAS and TWAS-related analyses in great detail, and the paper is well written. I have essentially no concerns with the analysis results which are all interesting. My main concern is that methodological innovation seems to be somewhat marginal (for example, several other methods including EpiXcan also incorporate functional priors) and so the manuscript would benefit from more detail on why exactly this method outperforms the state of the art (including outperforming fancy multi-tissue methods like UTMOST).

Major comments:

It's not entirely clear how this approach differs from EpiXcan, which also uses functional annotations to assign priors to the TWAS model weights. Why is PUMICE so much more effective than EpiXcan? Is it improved functional annotation data, or does PUMICE outperform EpiXcan even when the annotation data is the same? The work also shares some methodological similarities with PMID 29728367, which integrates enhancer/promoter interactions into TWAS model building and it would also be helpful to comment on how PUMICE differs from these two methods. Relatedly, I can see how better functional annotations or a more appropriate penalization can outperform non-functional single-tissue methods like PrediXcan, but the improvement versus UTMOST in Fig. 1c is hard to understand; UTMOST is essentially using data from thousands of additional cross-tissue samples, how is it possibly still underperforming? Some more intuition for why PUMICE wins would be very helpful to improve the longevity and impact of this work.

Line 179: The multi-tissue method comparison results do not appear to be provided. The relative

performance between PUMICE and other methods in independent data is a critical result and should be explored in detail. Can the authors comment on why PUMICE does so much better than UTMOST in-sample but not out-of-sample? Is this evidence that PUMICE is over-fitting to the GTEx data, given the large number of parameters and somewhat complicated nested cross-validation. Note that the UTMOST manuscript reported inflated accuracy due to improper cross-validation and over-fitting (as pointed out in PMC7606598) so this is not an uncommon error and important to rule out.

Figure 4a/TWAS comparisons: It's not clear to me what the precise figure of merit is here and whether it is significantly different between any of the methods? Moreover, the text assumes that λ_{GC} should be low, but inflated λ_{GC} can also occur under true polygenic heritable phenotypes (see PMID 21407268) and thus be an indicator of *better* overall performance. Should we interpret the higher λ_{GC} of UTMOST as the winning method, or focus on the tail effects, and why?

p8: I'm very puzzled by the choice of PUMICE+MASHR rather than PUMICE+UTMOST. In Supplementary Table 4, UTMOST achieves *substantially* better median correlation than MASHR, shouldn't this directly translate into improved TWAS statistics? Is it perhaps the case that MASHR captures models that PUMICE doesn't whereas UTMOST is redundant with PUMICE and, if so, can this be shown analytically?

Figure 4c: I think more explanation is needed for why DAPGW outperforms PUMICE in total number of TWAS associations even though PUMICE has a >300% gain in prediction accuracy ver DAPGW (Supp Table S3). If it's the case that DAPGW definitively produces inflated statistics then this should be explained more directly and I recommend moving the results from DAPGW (and any other inflated method) out of Figure 4 because they give the wrong impression.

Figure 4: Can the authors show that no other method + MASHR outperforms PUMICE+MASHR? This would also confirm that combining multiple models is not somehow inflating the overall statistics.

Minor comments:

I really appreciate the authors' attempts to investigate performance across a variety of metrics in Figure 4, it's clear that performance is complicated by redundancy and correlation of genes and the Figure does a thorough presentation of this without cherry picking.

Figure 4c/d: This isn't really about PUMICE, but I'm curious about why UTMOST is substantially underperforming even the single-tissue PrediXcan approach. This seems very different from what's presented in the UTMOST manuscript.

How important are 3D genome annotations versus standard epigenomic annotations (like this used in EpiXcan), chromatin conformation data can be difficult to obtain in certain cell-types and so it would be useful to know if it is critical to the success of this approach.

Figure 5: Is PUMICE+MASHR significantly better than UTMOST+MASHR in this analysis?

The COVID TWAS would benefit from a Manhattan plot figure, it's currently difficult to get a sense of how many more discoveries are being identified and where.

RESPONSE TO REVIEWER COMMENTS

Reviewer #1 (Remarks to the Author):

The authors propose a very interesting TWAS method to account for 3D genomic and epigenomic data. The application study results show promising performance by the proposed new method.

RESPONSE: Thank you for the positive and encouraging comments!

Here are my major concerns:

1. In the abstract, the authors write "Current approaches begin with a genomic interval of given size surrounding the gene start and end sites. This criterion lacks rigorous biological justification and hence results in less accurate models." I do not agree with this strong statement. Current approaches only consider eQTL within a specified window around gene start and end sites, which may show additional biological information can be used from additional 3D genomic and epigenomic data but not as the authors' statement of "lacks rigorous biological justification and results in less accurate models". As shown in the real application studies, the proposed method with a single tissue type won't outperform the multi-tissue TWAS method with a similar sample size. These results show the proposed method is comparable with some existing TWAS methods under certain situations.

RESPONSE: Thank you for the suggestion! We updated the wordings of those sentences as recommended, i.e., "Current approaches usually only consider eQTL within a specified window around gene start and end sites (i.e., 1 Mb window). Here we propose to integrate additional biological information (e.g., 3D genomic and epigenomic data) with eQTL to derive more accurate prediction models via an integrative method PUMICE (Prediction Using Models Informed by Chromatin conformations and Epigenomics)."

For tissues with a relatively small training sample size ($n < 100$), the multi-tissue TWAS method produces slightly more accurate gene expression prediction models than single-tissue TWAS methods and identifies more gene x trait associations. Yet, the identified associations by UTMOST tend to be copied across tissues, as UTMOST is based on group lasso method that preferentially select predictors shared across tissues. When comparing the number of unique gene x trait associations (which count gene x trait associations identified in multiple tissues only once), the median chi-square statistics at known associations, PUMICE outperforms UTMOST as we showed in the results. As training sample size gets to be medium and large, single tissue TWAS methods can outperform multi-tissue methods in term of prediction accuracy, the number of significant gene expression models, and TWAS power.

In addition, multi-tissue TWAS methods require datasets that measure gene expressions from multiple tissues of the same individuals (e.g., GTEx), and thus cannot effectively utilize much bigger single tissue datasets (e.g., CMC, DGN, GEUVADIS). Single tissue gene expression models

trained in CMC, DGN, or GEUVADIS can yield much higher TWAS power compared to multi-tissue models trained from GTEx (Figure 1, Supplementary Table 8).

2. Although the authors show "PUMICE resulted in an average of 30% and 16% improvement in imputation accuracy and an average gain of 52% and 12% more significant models" and "+ identified 51% more independent novel genes, increased mean chi-square statistics values at known loci by 26.5%, as well as achieved narrower credible interval for 73% of the traits being analyzed than the second-best method", the authors should also include simulation studies to compare gene expression prediction accuracy, TWAS power, and Type I error of the proposed method, with a comparison with existing methods.

RESPONSE: Thank you for the suggestion! We performed extensive simulations to compare gene expression prediction accuracy, TWAS power, and Type I error rates between PUMICE and existing TWAS methods. The results are concordant with comparisons using real data and showed improvements by PUMICE over other methods (Figure 1 and Supplementary Table 1-3). A detailed description of the simulation study design is shown in the Methods section.

3. It seems that the proposed method will do nested cross-validation to select the best window size for test SNPs. This is not clearly written in the main text. Since the test window size varies from gene to gene, tissue to tissue, comparing the credible interval size with the other TWAS methods that use 1MB region around gene start and end is not fair. Also, the discussion of the median number of test SNPs is also not fair.

RESPONSE: Thank you for the comment! Indeed, we used nested cross validation to select the best window size (and other tuning parameters). We have clarified this in the manuscript (Page 18, Section "Model Assessment in GTEx using cross-validation"). It is correct that the window size influences the size of the credible set, as PUMICE model has smaller number predictors from possibly smaller window sizes, and reduces the correlations between TWAS statistics of different genes. It thus can improve the fine-mapping accuracy and leads to smaller credible sets compared to TWAS methods that use fixed window sizes. In addition, adaptively choosing different window sizes helps prioritize causal variants and often yields better prediction accuracy for gene expression levels compared to methods that use a fixed window size.

4. How to define "independent" or "fine-mapping" significant gene is important and should be written clearly along with the motivation in the main text. How to define "significant" test gene should also be made clear in the main text.

RESPONSE: Gene x trait association is considered significant if the p-value is less than the Bonferroni threshold (0.05/the number of tests performed for each tissue) (Page 19, Paragraph 1). The p-value threshold used for each tissue can be found in Supplementary Table 25.

We first define locus and then define independent genes to be the top gene in each locus. Specifically, to define locus, we first rank all genes by their p-values from small to large. For the

top gene in the list, we define the locus as the window from -1Mb from gene start to +1Mb of gene end position, and any gene that overlaps the window is considered part of the locus. For all remaining genes outside the locus, we repeat the process described above and define another locus until there are no genes remaining with significant TWAS p-values (Page 9, Paragraph 6). We performed fine-mapping for genes in each locus (Page 20, Paragraph 1).

5. Descriptions about these 3D genomics informed regions and epigenetic information would be helpful in the main text to help readers to understand why these would be biologically relevant.

RESPONSE: We provided the descriptions of epigenomics and 3D genomics and highlighted their biological relevance in the Introduction section (Page 3, Paragraph 3).

6. The authors compared with many other TWAS methods besides PrediXcan and UTMOST, like FUSION and TIGAR. But only discussed the comparison results with respect to the number of significant test genes, median correlation values with PrediXcan and UTMOST. I think the authors should also include FUSION and TIGAR into the comparison.

RESPONSE: Thank you for the suggestion! We added the comparisons between PUMICE and FUSION/TIGAR (Page 6, Paragraph 4, Figure 1-2), and the advantage of PUMICE remains.

7. Since even nested cross-validation prediction results would be affected by overfitted models, I think the authors should focus more on comparing the gene expression prediction accuracy with independent data sets instead of focusing on discussing the performance within GTEx tissues.

RESPONSE: Thank you for the comment! For nested cross validation comparisons, although PUMICE shows better median Spearman's rank correlation coefficient than UTMOST, the difference is not statistically significant for tissues with small sample sizes. We agree that a multi-tissue method such as UTMOST yields better prediction Spearman's rank correlation coefficient when training sample size is small and identifies more gene x trait associations. On the other hand, the identified associations by UTMOST tend to be copied across tissues, as UTMOST is based on group lasso that preferentially select predictors shared across tissues. When comparing the number of unique gene x trait associations (which count gene x trait associations identified in multiple tissues only once) and the median chi-square statistics at known associations, PUMICE outperforms UTMOST as we showed in the results (Figure 5). As training sample sizes get to be medium or large, PUMICE outperforms UTMOST consistently. We provided complete comparisons of the gene expression prediction accuracy with independent datasets in Supplementary Table 6 and 8.

8. For gene expression prediction accuracy, comparing rank correlation might make more sense than person correlation.

RESPONSE: Thank you for the comment! We now use Spearman's rank correlation coefficient to measure the accuracy of gene expression prediction.

9. In our studies, we noticed using the Z-score test statistic as used in this paper with non-standardized SNP weights derived from non-standardized reference gene expression traits and genotype data will cause huge inflation. If the SNP weights were derived from non-standardized reference data in this paper, I would suggest using the S-PrediXcan test statistic for the gene-trait association study.

RESPONSE: Thank you for the comment! We confirmed that we indeed used standardized genotype data and centered expression data in our analyses.

10. Text labels in the figures are too small to read.

RESPONSE: Thank you for the comment! We updated all figures and made them more legible to read.

Reviewer #2 (Remarks to the Author):

This manuscript describes a new method, PUMICE, to integrate genomic/epigenomic information for TWAS analysis. PUMICE first divides SNPs into two groups that include an essential group and a non-essential group. PUMICE then places different penalties on the SNP effects in two groups separately for building the gene expression prediction models. The authors show that PUMICE outperforms many TWAS methods in both internal and external predictions, and performs well in TWAS and TWAS fine-mapping analysis across multiple traits as well as for drug repurposing.

Overall, the paper is well written and easy to follow. The proposed method is also new. However, the benefits of PUMICE for TWAS analysis are not yet demonstrated in the paper, as the other methods are not compared in a fair way with PUMICE (main comment #4 below). In particular, PUMICE does not outperform the other TWAS methods for TWAS applications across the complex traits and only PUMICE+MASHR does. However, PUMICE+MASHR is a simple strategy that combines PUMICE with MASHR and can be equally applied to all other TWAS methods for them to be combined with MASHR. Therefore, a thorough and fair comparison of PUMICE+MASHR with other TWAS methods + MASHR would be critical for evaluating the performance of PUMICE in the TWAS applications.

RESPONSE: Thank you for comment! In the revision, we decided to remove DAPGW and MASHR as the comparison can be confusing. In fact, as noted by Reviewer 3, the number of significant models using DAPGW and MASHR are inflated, as they use a fine mapping procedure in the entire sample to select predictors, which induces correlations between training and testing data. This is an acknowledged issue by the original article that proposed their uses to make prediction models [Barbeira et al., PMID: 32964524]. It makes it difficult and confusing to compare with other methods.

PUMICE consistently outperformed all remaining single tissue methods, i.e., EpiXcan, FUSION, PrediXcan, and TIGAR in terms of prediction accuracy, the number of significant models, and TWAS power, as we showed using simulation studies and applied data analyses.

PUMICE also outperforms UTMOST when training sample sizes are medium to large. UTMOST has slightly better prediction accuracy than PUMICE when training sample sizes are small. In applied data analysis, we noted that although UTMOST identified more gene x trait associations, these associations tend to be copied across tissues. This is due to the use of group lasso algorithm in UTMOST that preferentially select shared predictors across tissues, as acknowledged by authors of UTMOST. Compared to UTMOST, PUMICE identified more unique gene x trait associations (which count gene trait associations identified in multiple tissues only once) and the number of independent gene x trait associations, and produced larger mean chi-square statistics at known loci, all of which indicating improved TWAS power of PUMICE over UTMOST.

In addition, we also tried to combine each single tissue method with UTMOST using Cauchy combination p-values, and showed PUMICE+UTMOST outperformed other combined methods.

My main comments are listed below:

1. First, it is unclear how those non-penalization-based methods, including FUSION-BSLMM, TIGAR-VB, TIGAR-MCMC, were applied. Those methods can make use of the entire data, including both training and validation data, for model fitting. Consequently, those non-penalization-based methods can take advantage of the larger sample size in the combined training and validation data to improve prediction. It is important to fit those methods correctly and describe the fitting details in the methods section.

RESPONSE: Thank you for the suggestion! This is a wonderful point and we added the detailed description of the cross-validation process in the Methods section (Page 18, Section "Model Assessment in GTEx using cross-validation"), i.e., "We randomly split the data into five folds. For each iteration, we set 4/5 of the data as training fold and the remaining 1/5 as test fold. For methods that require parameter tuning (i.e., PrediXcan, EpiXcan, PUMICE, and UTMOST), we performed nested cross validation, where we further performed 5-fold cross-validation within training data to select tuning parameters that minimize mean squared error of the model. We then re-trained the model in the training fold (4/5 of the entire dataset) using selected tuning parameters and evaluated the performance in the testing fold (1/5 of the entire data) via Spearman's rank correlation coefficient (r). For methods that do not require parameter tuning (i.e., FUSION and TIGAR), we trained the model directly using training folds and evaluated the performance in the testing fold. We repeated this process five times. Final reported correlation coefficient was calculated based on the average of the five Spearman correlation estimates from the cross validations. To estimate the statistical significance of the prediction models, we followed the PrediXcan pipeline which uses Stouffer's method to convert averaged correlation to z-score and then to p-value."

2. In the external prediction section, the authors concluded that "The performance of single-tissue TWAS methods improves with bigger training sample size and often surpasses multi-tissue methods in GTEx tissues with large sample sizes" (line 187-188). However, the benefits of using single-tissue with larger sample size do not appear to be explored in the later TWAS analysis section, where the GTEx data but not the other three single-tissue data was used for TWAS analysis on the 82 complex traits. It would be important to apply various TWAS methods on DGN, CMC, and GEUVADIS for the TWAS analysis of the 82 complex traits. Would PUMICE still work better than the other TWAS methods when single-tissue expression data is used for TWAS analysis of these complex traits?

RESPONSE: Thank you for the suggestion! We now trained all single-tissue TWAS methods on CMC/DGN/GEUVADIS and applied prediction models to 79 GWAS complex traits. As expected, as training sample size increases, PUMICE outperforms the multi-tissue TWAS method

(Supplementary Table 9). We observed that PUMICE still achieved significantly higher median chi-squared values at MAGMA-prioritized loci in comparison to other single-tissue TWAS methods. The relative gain of the number of gene x trait associations by PUMICE, in comparison to other single-tissue TWAS methods, remains albeit decreases as training sample size increases (Supplementary Table 9). This is in agreement with what FUSION paper reported (i.e., power for TWAS does not substantially increase beyond 1,000 expression samples expression panel as the panel nearly saturates the available imputation accuracy).

3. PUMICE+MASHR requires the input of a p-value from MASHR. However, MASHR a Bayesian method and uses a local false sign rate (lfsr) to declare significance. How did you get a p-value from MASHR? Did you obtain a p-value from its Bayesian factor using Wakefield's method? If so, how do you justify the use of Wakefield's method, given that it only applies to a logistic regression model with one predictor variable?

RESPONSE: Thank you for the comment! We apologize for the confusion! Here, MASHR p-values refer to the statistical significance of gene expression prediction models using weights estimated from MASHR, i.e., if the correlations between measured and predicted gene expression is significantly different from 0. MASHR was used to jointly analyze multiple tissues and get a more accurate eQTL effect size estimate, which is then used to predict gene expressions.

As mentioned in response to Reviewer#3, the number of significant models and prediction accuracy using DAPGW and MASHR are inflated, as they use a fine mapping procedure in the entire sample to select predictors, which induces correlations between training and testing data. This is an acknowledged issue by the original article that proposed the use of them to make prediction models [Barbeira et al., PMID: 32964524]. It makes it difficult and confusing to compare with other methods. We decided to remove DAPGW and MASHR out of our analysis to avoid confusions in subsequent comparisons, as suggested by Reviewer 3.

4. PUMICE+MASHR is a simple strategy that combines the PUMICE p-value with the MASHR p-value. Such strategy can be easily applied to all other TWAS methods. To demonstrate the benefits of PUMICE, it would be important to compare PUMICE+MASHR with the performance of each other method + MASHR. For example, given that UTMOST yielded the lowest score for computational drug repurposing (CDR), would UTMOST+MASHR also work better than PUMICE+MASHR? Similarly, in Figure 4, DAPGW identified more unique genes than PUMICE and also fewer new genes, suggesting potentially fewer false signals and more power than PUMICE; would DAPGW+MASHR also outperform PUMICE+MASHR?

RESPONSE: As we explained in our response to the 3rd comment above, we have removed DAPGW and MASHR from the comparison to avoid confusions, as they yield inflated number of significant models. On the other hand, we sought to combine each single-tissue TWAS methods with UTMOST and confirmed that PUMICE+UTMOST is the most powerful method and outperformed all other combined methods.

5. For CDR, why examining only immune-related traits but not all the 82 traits? Does PUMICE perform better than the other methods for the other traits? I presume both positive and negative connectivity scores would be helpful for evaluating quantitative traits?

RESPONSE: Thank you for the suggestion! We now performed computational drug repurposing across all 23 diseases with known drug indications in the CLUE database and having at least 10 positively and 10 negatively associated genes (in at least 4 out of 9 TWAS methods) (Figure 6 and Supplementary Table 18). The remaining phenotypes are quantitative traits based on largely healthy individuals in population-based studies, such as height. It is unclear to us how to interpret drug repurposing results for normal trait variation.

6. For the fine-mapping results, it would be helpful to compare the performance of PUMICE directly with the standard TWAS fine-mapping method FOCUS. FOCUS is the standard TWAS fine-mapping method and relies on a fine-mapping algorithm that is different from the one described in the method section.

RESPONSE: Thank you for the suggestion. In the revision, we performed TWAS fine-mapping with FOCUS and compared the results with our fine-mapping method. The results can be found in Supplementary Table 20. It is clear that PUMICE leads to similarly improved fine-mapping resolution using FOCUS compared to other TWAS methods. We also found that our results are highly consistent with FOCUS, where 78% of the credible sets from FOCUS and our method contain the same genes.

Minor comments:

line 115: what is the parameter a ? Is this a typo? This parameter is not in the equation and not explained in the paragraph.

RESPONSE: Thank you for pointing this out. It is indeed a typo and we fixed this.

line 251: 2) -> 3)

RESPONSE: Thank you for pointing this out. We fixed this.

line 258: 3) -> 4)

RESPONSE: Thank you for pointing this out. We fixed this.

Reviewer #3 (Remarks to the Author):

The work presents a novel method for training gene expression prediction models for TWAS. The method uses penalized regression, with prior functional information on "essential" regions used to specify a different penalty for variants in these regions than other variants. The method is rigorously evaluated against a variety of alternative TWAS approaches and yields higher accuracy in-sample and in independent data (I commend the authors for seeking out diverse independent datasets to evaluate performance). When used for TWAS, the method seems to identify a larger number of unique/independent associations than any other single-tissue approach, and does best when also combined with a multi-tissue approach. A COVID TWAS and drug repurposing study is carried out, but it is difficult to evaluate the significance of these findings since we have so little ground truth about the phenotype or effective treatments. Overall this is a very thorough analysis that investigates many aspects of TWAS and TWAS-related analyses in great detail, and the paper is well written. I have essentially no concerns with the analysis results which are all interesting. My main concern is that methodological innovation seems to be somewhat marginal (for example, several other methods including EpiXcan also incorporate functional priors) and so the manuscript would benefit from more detail on why exactly this method outperforms the state of the art (including outperforming fancy multi-tissue methods like UTMOST).

Major comments:

1. It's not entirely clear how this approach differs from EpiXcan, which also uses functional annotations to assign priors to the TWAS model weights. Why is PUMICE so much more effective than EpiXcan? Is it improved functional annotation data, or does PUMICE outperform EpiXcan even when the annotation data is the same? The work also shares some methodological similarities with PMID 29728367, which integrates enhancer/promoter interactions into TWAS model building and it would also be helpful to comment on how PUMICE differs from these two methods. Relatedly, I can see how better functional annotations or a more appropriate penalization can outperform non-functional single-tissue methods like PrediXcan, but the improvement versus UTMOST in Fig. 1c is hard to understand; UTMOST is essentially using data from thousands of additional cross-tissue samples, how is it possibly still underperforming? Some more intuition for why PUMICE wins would be very helpful to improve the longevity and impact of this work.

RESPONSE: Thank you for your insightful comments!

There are several notable differences between EpiXcan and PUMICE:

1) PUMICE utilizes both epigenomic and 3D genomics data to create gene expression prediction models, while EpiXcan only uses epigenetic data. 3D genomic data can further provide information on genome organization and gene regulation and allows us to define biologically meaningful regions. We found that utilizing both epigenomic and 3D genomic data can effectively guide elastic nets to create better gene expression prediction models in general.

2) PUMICE is an adaptive method that allows us to choose regulatory regions and genetic variants adaptively. Given the complexity of gene regulation and still limited amount of epigenetic marks, it is important to accommodate the scenarios where regulatory regions defined by linear distance from the genes (e.g., 250kb, 1Mb windows) yield better performance. PUMICE in this regard, can be more advantageous compared to EpiXcan that rely on pre-computed weights.

Next, we added the comparison between PUMICE and EpiXcan when both methods use the same epigenetic annotations. Specifically, we provided PUMICE with the epigenomic annotation used by EpiXcan (i.e., Roadmap ChromHMM states). Using Roadmap annotation data, we found that PUMICE achieved 6.81% higher median Spearman's correlation than that of EpiXcan (Supplementary Table 7). Using SCREEN annotation data, we found that PUMICE achieved 8.17% higher median Spearman's correlation than that of EpiXcan (Supplementary Table 7). This further illustrated the flexibility of PUMICE to directly utilize various sources or formats of functional annotation data (unlike EpiXcan which needs to retrain for optimal Bezier equations).

We also clarified the difference between PUMICE and Wu and Pan [PMID: 29728367]. The work by Wu and Pan seeks to combine single variant association tests based on enhancer-promoter interactions information. It is not a TWAS method, but as the authors suggest, can complement the TWAS method. PUMICE, on the other hand, is a TWAS method, which uses epigenetic and 3D genome data to better predict gene expression levels and enhance association power (Page 13, Paragraph 3).

Finally, we clarified the comparison between UTMOST and PUMICE. As we clarified in the response to other comments, UTMOST borrows strengths across different tissues using group lasso, which forces the selection of the same variants as predictors across tissue types. This method improves power for tissues with small sample sizes, but may not fully capture between-tissue differences. This is an acknowledged issue by the authors of UTMOST ("By imposing this joint penalty across tissues, UTMOST encourages eQTL that are shared across tissues, but it still keeps tissue-specific eQTL with strong effects.") [Hu et al., PMID: 30804563].

In our comparison, we noted 1) UTMOST has some advantages for tissues with small sample sizes. But PUMICE quickly outperforms UTMOST for tissues with medium or large training sample sizes. 2) In TWAS, while UTMOST identified more gene x trait associations, many of these gene trait associations are copied across tissues. When comparing the number of unique gene x trait associations (i.e., gene x trait association identified in multiple tissues are counted only once), or the number of novel associations, PUMICE clearly outperforms UTMOST. 3) UTMOST requires datasets that measure multi-tissue gene expressions on the same individuals. For many tissues, bigger single-tissue datasets exist. When trained in bigger single tissue datasets, PUMICE models clearly outperform UTMOST models that are trained using GTEx data.

2. Line 179: The multi-tissue method comparison results do not appear to be provided. The relative performance between PUMICE and other methods in independent data is a critical result and should be explored in detail. Can the authors comment on why PUMICE does so much better than UTMOST in-sample but not out-of-sample? Is this evidence that PUMICE is over-fitting to the GTEx data, given the large number of parameters and somewhat complicated nested cross-validation. Note that the UTMOST manuscript reported inflated accuracy due to improper cross-validation and over-fitting (as pointed out in PMC7606598) so this is not an uncommon error and important to rule out.

RESPONSE: Thank you for the comment! We now provided the multi-tissue method comparison results in Supplementary Table 6 and 8. We described our cross-validation procedure in Methods (Page 18, Section “Model Assessment in GTEx using cross-validation”), which follows the same workflow as shown in UTMOST. We note that although PUMICE shows slightly better median Spearman’s correlation coefficient, the difference is not statistically significant for tissues with small sample sizes in cross-validation.

3. Figure 4a/TWAS comparisons: It's not clear to me what the precise figure of merit is here and whether it is significantly different between any of the methods? Moreover, the text assumes that λ_{GC} should be low, but inflated λ_{GC} can also occur under true polygenic heritable phenotypes (see PMID 21407268) and thus be an indicator of *better* overall performance. Should we interpret the higher λ_{GC} of UTMOST as the winning method, or focus on the tail effects, and why?

RESPONSE: Thank you for pointing this out! We agree with the reviewer that the genomic control value may either reflect inflation of type I errors or improved power, and hence is not helpful for comparing different methods. We have thus removed the comparison using the lambda genomic control value and removed it from the QQ plot.

4. p8: I'm very puzzled by the choice of PUMICE+MASHR rather than PUMICE+UTMOST. In Supplementary Table 4, UTMOST achieves *substantially* better median correlation than MASHR, shouldn't this directly translate into improved TWAS statistics? Is it perhaps the case that MASHR captures models that PUMICE doesn't whereas UTMOST is redundant with PUMICE and, if so, can this be shown analytically?

RESPONSE: Thank you for the comment! As other studies, we performed TWAS only for genes with significant gene expression prediction models. Yet, the number of significant gene prediction models are inflated for DAPGW and MASHR models, which makes it impossible to compare them with other models. Specifically, DAPGW and MASHR rely on a fine mapping procedure in the entire sample to select predictors, which induces correlations between training and test data and inflate the number of significant models and prediction accuracy. This is also mentioned in the original work by Barbeira et al (PMID: 32964524) that “there is no natural prediction performance measure in this scenario as cross-validation was not performed”. Furthermore, DAPGW and MASHR models tend to select a few SNPs (1-2 SNPs) in each gene

expression model, resulting in low prediction accuracy. As a result, we excluded the MASHR method from our analysis and only focused on combining single-tissue TWAS methods to UTMOST.

5. Figure 4c: I think more explanation is needed for why DAPGW outperforms PUMICE in total number of TWAS associations even though PUMICE has a >300% gain in prediction accuracy ver DAPGW (Supp Table S3). If it's the case that DAPGW definitively produces inflated statistics then this should be explained more directly and I recommend moving the results from DAPGW (and any other inflated method) out of Figure 4 because they give the wrong impression.

RESPONSE: Thank you for pointing out this important observation! As explained above in our response to comment 4, the number of significant models for DAPGW and MASHR is inflated. We have thus removed DAPGW and MASHR from the comparison.

6. Figure 4: Can the authors show that no other method + MASHR outperforms PUMICE+MASHR? This would also confirm that combining multiple models is not somehow inflating the overall statistics.

RESPONSE: As mentioned above, both DAPGW and MASHR yield inflated number of significant models, which are removed from the comparisons. However, we sought to combine each single-tissue TWAS method with UTMOST and confirmed that PUMICE+UTMOST outperformed all other combined methods.

Minor comments:

1. I really appreciate the authors' attempts to investigate performance across a variety of metrics in Figure 4, it's clear that performance is complicated by redundancy and correlation of genes and the Figure does a thorough presentation of this without cherry picking.

RESPONSE: Thank you for the positive and encouraging comments!

2. Figure 4c/d: This isn't really about PUMICE, but I'm curious about why UTMOST is substantially underperforming even the single-tissue PrediXcan approach. This seems very different from what's presented in the UTMOST manuscript.

RESPONSE: Thank you for the comment! Based on our observation, UTMOST has slightly better prediction accuracy when the training sample sizes are small. But PUMICE has better prediction accuracy as training sample sizes get to be medium and large. UTMOST borrows strengths from shared eQTL effects across tissues using group lasso, which preferentially selects the same variants as predictors across different tissues. It thus helps when training sample sizes are small, but mitigates tissue-specific differences, which is an issue acknowledged by the authors of UTMOST ("By imposing this joint penalty across tissues, UTMOST encourages eQTL that are shared across tissues, but it still keeps tissue-specific eQTL with strong effects.") [Hu et al., PMID:

30804563]. As a result, while it can identify many associated genes in TWAS, a large fraction of these associated genes are identical between tissues. When counting unique genes, UTMOST does not perform well. In addition, when the training sample size gets larger, the advantage of UTMOST diminishes, and does not perform as well as single tissue methods in TWAS.

3. How important are 3D genome annotations versus standard epigenomic annotations (like this used in EpiXcan), chromatin conformation data can be difficult to obtain in certain cell-types and so it would be useful to know if it is critical to the success of this approach.

RESPONSE: Thank you for the comment! In our analyses, for tissues without 3D genome data, we designated a proxy tissue with 3D genome data whose global gene expression profiles are similar based on clustering, as 3D genome structure is strongly correlated with gene expression profiles. We used the 3D genome data from the proxy tissue in the analysis. The performance of PUMICE is dependent on the resolution of the 3D genome organizations (e.g., loop, TAD), which in turn depends on the read depth of Hi-C experiments. Importantly, we noticed that the performance of PUMICE trained in lymphoblastoid cell line data is considerably better (according to the external prediction accuracy) and this in part could be due to the very high read depth of Hi-C data available for lymphoblastoid cell lines (Supplementary Table 6). As numerous efforts are underway to generate high resolution 3D genome data (e.g., with Micro-C), we expect that methods that integrate 3D genomes for modeling gene expressions will continue to improve.

4. Figure 5: Is PUMICE+MASHR significantly better than UTMOST+MASHR in this analysis?

RESPONSE: As mentioned above in the response to major comment 4, DAPGW and MASHR had inflated number of significant models. We thus decided to remove DAPGW and MASHR out of our analysis, as it is not possible to properly evaluate the number of significant models. However, we sought to combine each single-tissue TWAS method to UTMOST and confirmed that PUMICE+UTMOST is the best and outperformed all other methods based on combined p-values.

5. The COVID TWAS would benefit from a Manhattan plot figure, it's currently difficult to get a sense of how many more discoveries are being identified and where.

RESPONSE: Thank you for the suggestion! We added a Manhattan plot for COVID TWAS results (Figure 7).

REVIEWER COMMENTS

Reviewer #1 (Remarks to the Author):

The authors have well addressed my comments. I only have two minor comments:

1. The Manhattan plot in Figure 7 might be plotted as one per phenotype in a separate panel. The current figures look quite confusing.
2. Lots of the complementary tables seem to contain information that has been plotted in the complementary figures, or could be plotted as Bar plots or box plots to better represent/visualize the results. Having those complicated tables, some of which are of hundreds of rows, makes it hard for the readers to process the information.

Reviewer #2 (Remarks to the Author):

Most of my previous comments were well addressed. I have a few remaining comments for the simulation part that was added during the revision.

1. The simulations are not realistic. Specifically, $h_p^2=0.1$ is a huge number and it is hard to find a gene that would explain 10% of phenotypic variance for a complex trait. Similarly, $h_e^2=0.1$ is also a large number, though not as extreme as $h_p^2=0.1$. It would be important to examine type I error control and power for different methods at realistic h_p^2 and h_e^2 values that are way below 0.1.
2. Similarly, are the used epigenomic enrichment factors (EF) in the simulations realistic? What are the estimated the epigenomic enrichment factors in the real data?
3. The type I error shown on Fig 1a looks concerning. It looks like all methods produce inflated type I error well at the cutoff of 0.05. Do these methods control for type I error well at smaller p-value cutoffs? It would be important to show qq-plots for these null simulations.

Reviewer #3 (Remarks to the Author):

The authors have addressed all of my comments in the revisions.

It is now more clear where and why PUMICE outperforms the competing methods, and in particular the gains versus EpiXcan observed in independent data provide very helpful context: PUMICE outperforms EpiXcan a little in terms of median correlation (1-10% depending on the tissue; Supp. Table 6) while identifying many more significant genes (Supp. Table 7). Given that completely independent data is the gold standard for validation, I recommend including these PUMICE vs EpiXcan results as a main figure sub-panel.

One other minor comment regarding line 265: "Overall, PUMICE outperforms all other methods (i.e., EpiXcan, PrediXcan, TIGAR, FUSION, and UTMOST) ... (Supplementary Fig. 6 and Fig. 5)". This sentence mentions that PUMICE outperforms EpiXcan but I do not see the EpiXcan results in Supp Fig. 6 or Fig. 5. My understanding is that EpiXcan was not included in the analysis of 79 complex traits because too few EpiXcan models/tissues were available but it would be helpful to clarify here and omit EpiXcan from the parenthetical if EpiXcan was not compared to (or perhaps the authors were referring to Supp Table 9?). I also did not see a reference to Supp Table 10 in the main text.

These are both minor/stylistic suggestions and do not require re-review.

RESPONSE TO REVIEWERS

Reviewer #1 (Remarks to the Author):

The authors have well addressed my comments. I only have two minor comments: 1. The Manhattan plot in Figure 7 might be plotted as one per phenotype in a separate panel. The current figures look quite confusing.

RESPONSE: Thank you for the suggestion! We have separated the figure into 4 subplots (as shown below).

2. Lots of the complementary tables seem to contain information that has been plotted in the complementary figures, or could be plotted as Bar plots or box plots to better represent/visualize the results. Having those complicated tables, some of which are of hundreds of rows, makes it hard for the readers to process the information.

RESPONSE: Thank you for the suggestion! We converted these tables into **Supplementary Figure 1**.

Reviewer #2 (Remarks to the Author):

Most of my previous comments were well addressed. I have a few remaining comments for the simulation part that was added during the revision.

1. The simulations are not realistic. Specifically, $h_p^2=0.1$ is a huge number and it is hard to find a gene that would explain 10% of phenotypic variance for a complex trait. Similarly, $h_e^2=0.1$ is also a large number, though not as extreme as $h_p^2=0.1$. It would be important to examine type I error control and power for different methods at realistic h_p^2 and h_e^2 values that are way below 0.1.

RESPONSE: Thank you for this insightful comment. We redid the simulation with more realistic values for parameters (h_2e , h_2p), including (0.100, 0.005), (0.050, 0.010), and (0.025, 0.020). We show the new results in **Figure 1** and **Supplementary Figure 1**.

2. Similarly, are the used epigenomic enrichment factors (EF) in the simulations realistic? What are the estimated the epigenomic enrichment factors in the real data?

RESPONSE: Thank you for pointing this out. Our original values for epigenomic enrichment factors were concordant with results from Finucane et al (**Figure below**). For example, they found that enhancer, promoter, and DNase I hypersensitive site (DHS) regions showed enrichments of ~ 3.8 (with 6.3% of SNPs explaining an estimated 23.8% of SNP heritability), ~ 2.8 (with 3.1% of SNPs explaining an estimated 8.7% of SNP heritability), and ~ 1.7 (with 16.8% of SNPs explaining an estimated 28.5% of SNP heritability), respectively. In our new simulations, we used the epigenomic enrichment factors of 2 and 4, which closely match the real data.

Reference: Finucane, H., Bulik-Sullivan, B., Gusev, A. *et al.* Partitioning heritability by functional annotation using genome-wide association summary statistics. *Nat Genet* **47**, 1228–1235 (2015). <https://doi.org/10.1038/ng.3404>

3. The type I error shown on Fig 1a looks concerning. It looks like all methods produce inflated type I error well at the cutoff of 0.05. Do these methods control for type I error well at smaller p-value cutoffs? It would be important to show qq-plots for these null simulations.

RESPONSE: Thank you for pointing this out! In our new simulations, we increased the number of replicates to more accurately estimate type I errors under realistic parameter settings. We showed type I error at different p-value cutoffs in **Supplementary Table 1**, which are all well controlled. We also illustrated quantile-quantile plots under null hypothesis in **Supplementary Figure 2**, which shows that the global distribution for the p-values is well-calibrated.

Reviewer #3 (Remarks to the Author):

The authors have addressed all of my comments in the revisions.

It is now more clear where and why PUMICE outperforms the competing methods, and in particular the gains versus EpiXcan observed in independent data provide very helpful context: PUMICE outperforms EpiXcan a little in terms of median correlation (1-10% depending on the tissue; Supp. Table 6) while identifying many more significant genes (Supp. Table 7). Given that completely independent data is the gold standard for validation, I recommend including these PUMICE vs EpiXcan results as a main figure sub-panel.

RESPONSE: Thank you for the suggestion. We added these results in **Supplementary Figure 8**.

One other minor comment regarding line 265: "Overall, PUMICE outperforms all other methods (i.e., EpiXcan, PrediXcan, TIGAR, FUSION, and UTMOST) ... (Supplementary Fig. 6 and Fig. 5)". This sentence mentions that PUMICE outperforms EpiXcan but I do not see the EpiXcan results in Supp Fig. 6 or Fig. 5. My understanding is that EpiXcan was not included in the analysis of 79 complex traits because too few EpiXcan models/tissues were available but it would be helpful to clarify here and omit EpiXcan from the parenthetical if EpiXcan was not compared to (or perhaps the authors were referring to Supp Table 9?). I also did not see a reference to Supp Table 10 in the main text.

RESPONSE: Thank you for catching these mistakes! They are indeed typos, and we fixed all these issues.

These are both minor/stylistic suggestions and do not require re-review.

REVIEWERS' COMMENTS

Reviewer #2 (Remarks to the Author):

All my previous comments were well addressed.